civil engineering/engineering geology

blue brick, ancient city of Ping Yao, weathering, firing temperature, salt crystallization, frost resistance

**Author for correspondence:**
Zhong-jian Zhang
e-mail: zhangzhongjian@cugb.edu.cn

# Characteristics and weathering mechanisms of the traditional Chinese blue brick from the ancient city of Ping Yao

## Jian-bin Liu and Zhong-jian Zhang

Department of Civil Engineering, School of Engineering and Technology, China University of Geosciences (Beijing), No. 29 Xueyuan Road, Haidian District, Beijing 100083, People's Republic of China

Z-jZ, 0000-0002-5527-1006

The traditional blue brick was the dominant clay brick used in Chinese architecture before the mid-nineteenth century. The ancient city of Ping Yao, a United Nations Educational, Scientific and Cultural Organization (UNESCO) heritage site, is an outstanding example of blue brick architecture. The Ping Yao bricks within the damp areas (up to 4 m at highest) of the ancient city's walls and private houses are subjected to various weathering, including contour scaling, flaking, powdering and salt crystallization. This study aims to characterize the properties, analyse weathering mechanisms, determine the main weathering factors and discuss the anti-weathering strategies of blue bricks. To do so, samples of brick and salt efflorescence were collected from the historical buildings of Ping Yao and were studied with regard to their mineralogical and physico-mechanical (e.g. density, porosity, pore size distribution, water absorption and uniaxial compressive strength) properties. The resistance to salt crystallization and frost, and the maximum firing temperatures of the brick samples were determined in the laboratory. Weathering mechanisms and anti-weathering strategies were discussed. Salt crystallization and freeze–thaw cycles were found to be two important factors that lead to brick weathering. An anti-weathering strategy of 'damp blocking, desalination and brick replacing' was discussed based on the laboratory experiments, suggestions in literature and site conditions.

## 1. Introduction

Clay bricks have been widely used for structural and ornamental purposes in China for over 2000 years. The traditional blue brick

was the dominant clay brick used in both official and private buildings before the mid-nineteenth century. Since then, the blue brick has been gradually replaced by European red bricks and other modern products such as Portland cement [1].

The blue brick can be used as a construction material for walls, floors, roofs and balusters or as a decorative material for dwellings, temples and gardens. The brick can be used alone or in combination with stone, timber and clay. In China, there are many cultural relics constructed from traditional blue bricks. During the last decade, the properties and weathering of blue bricks have drawn increasing attention owing to the increased focus on conservation of cultural relics that have come with economic developments and a greater awareness of the importance of cultural protection.

Most existing studies have focused on red bricks and have achieved considerable results regarding the influence of the weathering factors and mechanisms; such as water impregnation during rainy seasons and water evaporation during dry seasons [2], the raw clay mineral compositions and the mineralogical and textural evolution during the firing process [3–5], the physical and mechanical properties [6], as well as the weathering behaviour [7]. Compared with the widely studied red bricks, the scientific understanding of the blue bricks have not received enough attention, even though the brick-making methods of the traditional Chinese blue brick are very different from the methods used in making European red bricks and thus the properties may be different [1].

To create Chinese blue bricks, clay and water need to be blended in exactly the correct proportions, formed into cuboid adobe structures and dried under natural conditions. Then, the adobes need to be fired in a sealed kiln to form the final product. The fuel used in the kiln is biomass fuel, such as wheat bran, fire wood or charcoal. When oxygen is abundant during the firing process, $Fe^{2+}$ in the adobes will be oxidized into $Fe^{3+}$, which gives the bricks a red or orange colour. When oxygen is insufficient or when water is added during the latter stage of the firing process, the biomass fuel and water will produce carbon monoxide or hydrogen, both of which can reduce $Fe^{3+}$ into $Fe^{2+}$, thereby forming a blue brick. The firing temperatures, which directly affect the textural evolution, vitrification, pore size distribution (PSD) of the bricks, and thus the physico-mechanical properties and weathering resistance, also show some differences between red and blue bricks according to the literature [1,3–5,7]. For example, Elert *et al*. [5] studied the influence of firing temperature on bricks made by two typical clays, and found that the low-temperature-fired (700–800°C) bricks generally present inferior PSD, strength, and weathering resistance including freeze–thaw and salt crystallization, compared with the high-temperature-fired (over 1000°C) bricks. López-Arce *et al*. [4] found that the firing temperature is directly related to bricks' properties by controlling the phase changes of minerals, e.g. the new mineral phases (gehlenite and diopside) were formed at the expense of calcite and dolomite above 800°C. Cultrone *et al*. [7] found the bricks used for the construction of the 'Triangul Bastion', Riga (Latvia) were fired at around 900°C; they also found the difference in colour corresponds to the difference in minerals: yellow colour bricks were characterized by quartz, diopside and minor amounts of feldspar; the red bricks were much richer in quartz and feldspar. The data regarding firing temperatures of Chinese blue bricks are limited in the literature, Shu *et al*. [1] studied 14 blue and red brick samples derived from four modern buildings (constructed between 1866 and 1935) in Shanghai, China and reported that the maximum firing temperature of blue bricks ranges from 600 to 800°C. The distinctions of brick-making environments (i.e. oxygen, temperature) should produce the blue bricks' unique characteristics that are worth investigating.

Given the differences existing between red and blue bricks in the raw materials and technical processes employed in their production, as well as the respective environments of the brick masonry in which they appear, it is necessary to investigate the characteristics and the weathering resistance of blue bricks and extend this understanding into the current knowledge base for bricks. For this reason, ancient blue brick samples collected from the ancient city of Ping Yao in Shanxi Province, China, were characterized in detail with regard to their mineral composition, physical and mechanical properties, resistance to freeze–thaw cycles and salt crystallization cycles. This paper presents the results of the phase one work of our study. The objective of this paper is to analyse and discuss the weathering factors of blue bricks and to propose preventive conservation methods based on laboratory experiments and field conditions, and to provide a relatively narrow range of methods to be further tested in the field. As there are many historic Chinese architectural structures constructed of blue brick, this study not only provides understanding and possible approaches on the conversation of brick masonry in the ancient city of Ping Yao, which is of great importance, but also relates the possible implications of those results to other similar cases.

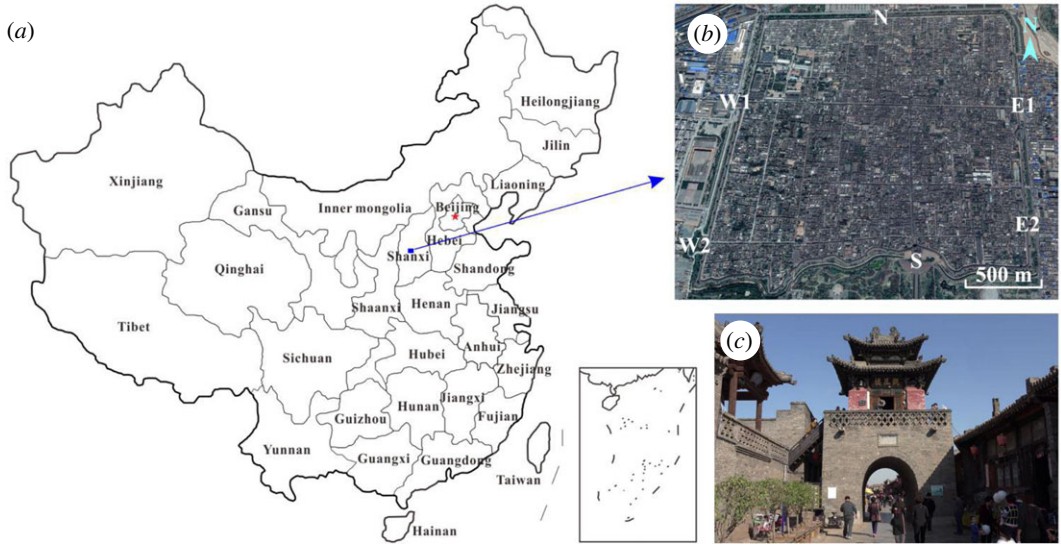

**Figure 1.** Location of the study area: (*a*) location of the ancient city of Ping Yao; (*b*) remote sensing image of the ancient city, where N is the north gate, S is the south gate, W1 and W2 are the two west gates, and E1 and E2 are the two east gates; and (*c*) a street inside the ancient city.

## 2. Study area

### 2.1. Brief introduction of the ancient city of Ping Yao

Located in Ping Yao County, central Shanxi Province (figure 1), the ancient city of Ping Yao is an exceptional example of a traditional Chinese city. It is the most complete existing ancient building complex in China, and was included in the World Culture Heritage List in 1997 because of its cultural value and state of preservation. Walls in the ancient city of Ping Yao are almost completely constructed from blue brick.

This ancient city was established in the King Zhouxuan era (828–738 BC) of the Western Zhou Dynasty. The city wall was originally made of packed earth. For military defence purposes, the original city wall was expanded with bricks starting from 1370 during the Ming Dynasty (1368–1644). The city buildings currently surviving were also mainly built during a similar period and, since then, have been restored or reconstructed. The city wall (10 m high and 6.1 km long) has a square layout with the central axis 10–15° north by west. This ancient city with six gates contains 4000 ancient private houses, 400 of which have great preservational value.

The environmental information provided by the Ping Yao Meteorological Bureau indicates that the average annual insolation reaches 2433.2 h. The average annual temperature is 10.4°C, ranging from −5.4°C in January to 24.2°C in July. The temperature generally below 0°C from mid-October to mid-April the following year. The average rainfall is 415.5 mm, falling mainly during the summer months. The rainfall from July to September accounts for 58.8% of the total annual rainfall amount, while the rainfall from December to February only accounts for 3.5% of the total annual rainfall. The annual relative humidity is 58%. The average wind speed is 2.2 m s$^{-1}$, being higher in spring and winter and lower in summer and autumn. The prevailing wind is mainly from the southwest but from the northeast between June and September.

### 2.2. Weathering types

Blue bricks in most walls of the ancient city of Ping Yao have suffered weathering (figure 2). The weathering is mainly distributed within the zone of rising damp, which is normally 1–3 m above the ground but can extend to 4–5 m in extreme cases. According to the International Council on Monuments and Sites-International Scientific Committee for Stone (ICOMOS-ISCS) [8] classification for general masonry heritages, the main weathering types include contour scaling, flaking, powdering and salt crystallization. The mortar (as indicated by arrows in figure 2*c*) in Ping Yao is mainly lime or a soil–lime mixture. The organic additive is found in important parts such as city walls. The type of organic additive used in Ping Yao is a thin gruel made from sticky rice and water.

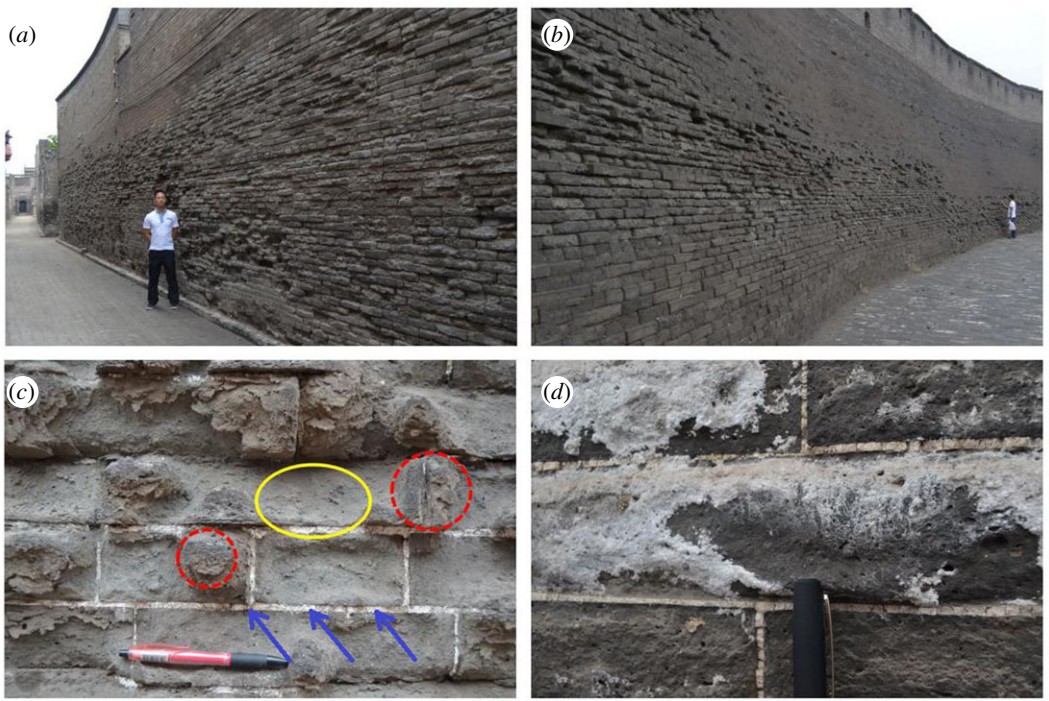

**Figure 2.** Weathering phenomena in the ancient city of Ping Yao: (*a*) the external walls of private houses, (*b*) the outer side of the city wall, (*c*) weathering of bricks and mortar, and (*d*) salt efflorescence on the surface area of bricks. The yellow ellipses and red circles stand for highly weathered concave parts and slightly weathered convex parts of the bricks, respectively. The mortar is indicated by arrows.

## 3. Experimental materials

Samples used in the study came from multiple locations of the ancient city of Ping Yao (table 1). Three types of samples were collected from different locations of the ancient city:

(i) weathered ancient brick samples, numbered PY-1 to PY-30, were taken from multiple dispersed locations among the walls of private houses and the city walls. The weights of the samples range from 10 to 50 g. The samples can be divided into two sub-types: soft, highly weathered powder samples from the concave parts of the bricks (yellow ellipses in figure 2) and hard, slightly weathered fragment samples from the convex parts of the bricks (red circles in figure 2). These samples were originally located within the influence range of rising damp (normally at around 0.5–1.0 m above the ground). The objective of this test was to investigate the possible causes of different weathering resistance in different parts of the same brick from perspectives of mineralogy, therefore, the mineral compositions of the samples were also determined;

(ii) salt efflorescence samples, numbered S-1 to S-3, were taken from the walls of private houses. The salt efflorescence samples were mainly used to determine the salt crystal types in the Ping Yao ancient city. Because different types of salt have different expansibility, thus the detrimental weathering effects to the brick are variant. Mineral compositions of the samples were determined; and

(iii) intact blue bricks were collected from five different locations in Ping Yao and were numbered '*i*-*j*', where '*i*' denotes the sampling location and '*j*' denotes the sequential number of samples at each location. The sampling locations 1–5 denote the locations of 'Yard No. 24, Xihujing Street', 'Luji Alley', 'Duanzhai Minsu Inn, Yard No. 21, Xinbu Alley, Beihai Street', 'Yunzhou Inn, Yard No. 2, Jiaochang Street', and 'west city wall and east city wall', respectively. These bricks were collected from restoration or reconstruction sites and were comparatively well preserved. Having had the surface removed, the bricks were cut into $4 \times 4 \times 4$ cm$^3$ cubic samples and numbered '*i–j–k*', where '*k*' denotes the sequential number of cubes cut out of each brick. The cubic samples are termed as *compact* samples in this paper. These samples were used for the physical, mechanical, maximum firing temperature, salt crystallization and freeze–thaw tests. Samples used to determine PSD were drilled out of these cubes.

**Table 1.** Blue brick samples used in the study. (Note that the '*i-j-k*' symbol is adopted here for the convenience of distinguishing different samples in the following sections.)

| name | sample ID | sum | shape | sampling location | testing item |
|---|---|---|---|---|---|
| fragment samples | PY-1 to PY-30 | 30 | fragments or powder | walls of the private houses and city walls | mineral compositions |
| salt efflorescence | S-1 to S-3 | 3 | powder | walls of the private houses in Ying'er Alley, Helan Bridge and Renyi Street | mineral compositions |
| intact blue bricks | *i–j–k* (e.g. 1–1–2) | 22 (cut into 86 samples) | intact or partially intact cuboids | walls of private houses and city walls | physical properties, firing temperature, salt crystallization, freeze–thaw cycles, uniaxial compressive strength, PSD |

# 4. Experimental methods

## 4.1. Mineralogical tests

The mineralogical understanding of building materials is indispensable to studying the preservation and restoration of architectural heritage sites [9]. The mineral compositions of the blue bricks were identified by powder X-ray diffraction (XRD) using an X-ray diffractometer (D/max-2500/PC, Rigaku, Japan) with Cu K$\alpha_1$ radiation ($\lambda = 1.54$ Å). The following conditions were used for the test: a voltage of 35 kV, a current intensity of 50 mA, an explored $2\theta$ range between 2° and 70°, a step size of 0.020° (1.2 s) and a scan speed of 1.000° min$^{-1}$. Each sample was ground separately until all of the particles passed through a sieve with 0.063 mm mesh.

## 4.2. Physico-mechanical properties test

### 4.2.1. Density, porosity and water absorption test

The apparent density ($\rho_b$), real density ($\rho_r$), open porosity ($p_o$) and total porosity ($p$) were determined using the Archimedes method and the pycnometer method in accordance with SL 264 [10] and BS EN 1936 [11] and calculated as $\rho_b = (m_d/(m_s - m_h)) \times \rho_{rh}$, $\rho_r = (m_e/(m_e - (m_1 - m_2))) \times \rho_{rh}$, $p_o = (m_s - m_d)/(m_s - m_h)$, and $p = (1 - \rho_b/\rho_r)$, respectively. The parameter $m_s$ is the mass of the saturated sample; $m_h$ is the mass of the sample immersed in water; $m_d$ is the mass of the dry sample; $\rho_{rh}$ is the density of water (0.998 g cm$^{-3}$ at 20°C); $m_1$ is the mass of the pycnometer filled with water and the ground sample; $m_2$ is the mass of the pycnometer filled with water, and $m_e$ is the mass of the ground and dried sample.

The free water absorption ($\omega_a$) and the forced water absorption ($\omega_s$) were determined by total immersion under atmospheric pressure and vacuum conditions, respectively, and these characteristics were calculated as $\omega_a = (m_w - m_d)/m_d$ and $\omega_s = (m_s - m_d)/m_d$, where $m_w$ is the mass of the sample after immersion in water at atmospheric pressure.

The saturation coefficient $C_s$ is the quotient of $\omega_a$ and $\omega_s$ [12]. This coefficient describes how much of the total pore space is accessible to water absorption, providing a value for the frost resistance evaluation [12,13].

### 4.2.2. Mercury intrusion test

The PSD of the bricks was determined by a mercury intrusion porosimeter (PoreMaster 60, Quantachrome Corporation, USA) with a maximum injection pressure of 60 000 psi. Five cylindrical samples (9 mm in diameter and 20 mm in height) drilled out of compact samples from five different locations were oven-dried for 24 h at 105°C and analysed.

### 4.2.3. Coefficient of water absorption by capillarity test

The coefficient of water absorption by capillarity is a parameter that characterizes the water-absorbing ability of masonry materials. Testing procedures were in accordance with British standard BS EN 1925 [14]. The dried sample was first placed in a tank into which deionized water (20°C) was then slowly introduced. Counting from the moment when the sample touched water, the sample was taken out for weight measurements at different time intervals (unit: s). The data were projected onto a Cartesian coordinate system, where $Y$ is the water absorption in g m$^{-2}$ and $X$ is the square root of time in s$^{0.5}$. The slope of the skew line is the coefficient of water absorption by capillarity. The test was carried out on nine dried cubic brick samples with side lengths of 40 mm.

### 4.2.4. Compressive strength test

The compressive strength test is inevitably the most reliable means to determine the uniaxial compressive strength (UCS) of masonry materials [15]. The UCS was carried out in a TAW-2000 electric-fluid servo-controlled testing system at a loading rate of 0.5 MPa s$^{-1}$ (GB/T 50266–2013) [16]. In the test, 10 compact dry samples (oven-dried at 105°C for 24 h) were used in the form of a cube with dimensions of $40 \times 40 \times 40$ mm$^3$.

## 4.3. Firing temperature test

During the firing process, the clay minerals in the brick adobes experienced characteristic reactions such as dehydroxylation, decomposition and transformation, which led to changes in the adobes' mineral phases and magnetic properties. Once the adobes are cooled from the maximum firing temperature, the processes of the neoformation of minerals will normally not be reversed. Therefore, the maximum firing temperature of the ancient bricks can be determined from the magnetic susceptibility variation curves after re-heating the samples [17].

The maximum firing temperature of the Ping Yao blue brick was assumed to range from 260 to 1000°C. The brick samples were re-fired at the lowest assumed temperature (i.e. 260°C), which was then held for 24 h. Then, the oven was switched off. The samples were collected after another 24 h and subjected to magnetic susceptibility measurements on a SM-30 magnetic susceptibility metre. This procedure was repeated in a 20°C increment step up to 1000°C. The maximum firing temperature can thus be determined from the curve of magnetic susceptibility as a function of the re-firing temperature. To read the firing temperature more accurately, the square of the first derivative was plotted as a function of the re-heating temperature [1,17].

## 4.4. Resistance to salt crystallization test

A salt crystallization test can provide information on the damaging effects of soluble salts that exist in the brick. In the experiment, ancient bricks were separately placed in 14% (mass percentage concentration) sodium sulfate ($Na_2SO_4$) solution, magnesium sulfate ($MgSO_4$) solution and sodium carbonate ($Na_2CO_3$) solution. Among these salts, $Na_2SO_4$ is considered one of the most dangerous because of its strong crystallization pressure in the pores and fissures [18,19]. $MgSO_4$, which was detected on the surface of ancient bricks, exhibits strong expandability during crystallization [20]. The expansion during the process of $Na_2CO_3$ crystallization is relatively weak [21].

In the test, brick samples were used with a size of $40 \times 40 \times 40$ mm$^3$ according to the standard of BS EN 12370 [22] recommendation. The dry samples were impregnated with salt solution for 2 h at 20°C and then heated for 16 h at 105°C. After heating, the samples were allowed to cool for 2 h at room temperature before being subjected to the next immersion phase. The cycles were repeated 15 times or until the sample failed. Nine samples were immersed in each solution. A total of 27 samples were used in the test.

## 4.5. Frost resistance test

The test was in accordance with the BS EN 12371 [23] recommendation. Cubic samples with a side length of 40 mm were dried at 70°C to a constant mass and cooled. The dried samples were placed upright in a container at least 15 mm from adjacent samples. Deionized water (20°C) was first introduced into the container until it reached 20 mm. Then, water was added up to 30 mm after 1 h and finally reached 65 mm after 2 h. After 48 h of immersion, the samples were taken out to freeze at −20°C for 12 h and then thawed in deionized water at 20°C for 2 h.

The test was carried out on nine cubic samples until visible damage occurred. The dry mass ($m_d$), free water absorption ($\omega_a$) and forced water absorption ($\omega_s$) were tested after every five cycles.

# 5. Results and analyses

## 5.1. Mineral compositions

The XRD results of the brick and salt efflorescence from the ancient city of Ping Yao are presented in table 2. The main minerals in Ping Yao ancient bricks (i.e. quartz and feldspar) are similar to those of ancient red bricks in Thailand [2], Spain [4] and Latvia [7].

Greater amounts of gypsum and calcite are found in weathered samples than in compact samples, especially in samples from concave parts. Gypsum is the product of the reaction of sulfur compounds in polluted atmospheres (e.g. $Ca^{2+} + SO_2 \rightarrow CaSO_4$) [24,25]. The clay mineral content in samples from greatest to least is concave parts > convex parts > compact samples, while the quartz content levels are in the reverse order. The difference in clay and quartz content levels across bricks is probably owing to discrepancies in raw clay minerals and brick-making techniques (e.g. uniformity of clay mixture,

**Table 2.** XRD results (wt%) of the ancient bricks from the ancient city of Ping Yao. (Concave and convex parts of the bricks denote the areas in the yellow ellipses and red circles in figure 2, respectively.)

| sampling location | sample no. | quartz | K-feldspar | albite | calcite | halite | gypsum | clay minerals |
|---|---|---|---|---|---|---|---|---|
| concave parts of the bricks | PY-1 | 51 | 9 | 20 | / | / | 1 | 19 |
| | PY-7 | 61 | 12 | 12 | / | / | 3 | 12 |
| | PY-9 | 38 | 5 | 10 | 17 | 5 | 1 | 24 |
| | PY-10 | 57 | 10 | 10 | 7 | / | 2 | 14 |
| | PY-16 | 57 | 9 | 17 | 6 | 2 | / | 9 |
| convex parts of the bricks | PY-3 | 61 | 17 | 13 | / | / | 1 | 8 |
| | PY-4 | 62 | 8 | 20 | / | / | 1 | 9 |
| | PY-13 | 58 | 20 | 19 | / | / | 1 | 2 |
| | PY-28 | 60 | 16 | 17 | 3 | / | 1 | 3 |
| compact bricks | 1-3 | 65 | 17 | 16 | / | / | / | 2 |
| | 4-1 | 65 | 16 | 13 | 4 | / | / | 2 |
| salt efflorescence | S-1 | hexahydrite: 59, epsomite: 39, gypsum: 2; | | | | | | |
| | S-2 | hexahydrite: 2, epsomite: 98; | | | | | | |
| | S-3 | sodium nitrate: >99, bloedite: <1. | | | | | | |

**Table 3.** Density, porosity, water absorption, saturation coefficient of the compact Ping Yao ancient bricks.

| parameters | mean value | standard deviation | sum of the samples |
|---|---|---|---|
| apparent density ($\rho_b$, g cm$^{-3}$) | 1.562 | 0.095 | 10 |
| real density ($\rho_r$, g cm$^{-3}$) | 2.779 | 0.047 | 10 |
| open porosity ($p_o$, %) | 37.97 | 2.87 | 10 |
| total porosity ($p$, %) | 43.78 | 3.30 | 10 |
| free water absorption ($\omega_a$, %) | 18.40 | 2.33 | 20 |
| forced water absorption ($\omega_s$, %) | 23.55 | 3.04 | 20 |
| saturation coefficient ($C_s$) | 0.78 | 0.03 | 20 |

local sintering temperatures). The inhomogeneities in the mineral composition may be a factor that results in different weathering resistances both among different bricks and also within different parts of the same brick (figure 2) [26].

Salt efflorescence on ancient bricks is mainly composed of hexahydrite ($MgSO_4 \cdot 6H_2O$), epsomite ($MgSO_4 \cdot 7H_2O$) and sodium nitrate ($NaNO_3$). Small amounts of gypsum ($CaSO_4 \cdot 2H_2O$) and bloedite ($Na_2Mg(SO_4)_2 \cdot 4H_2O$) are also found. Sulfate (e.g. magnesium sulfate and sodium sulfate) often presents strong expandability during crystallization [26].

## 5.2. Physico-mechanical properties of intact bricks

### 5.2.1. Density, porosity and water absorption

The density, porosity and water absorption of compact samples are listed in table 3.

The apparent density of the ancient brick is 1.562 g cm$^{-3}$, which is generally smaller than that of ordinary rocks [27]. Correspondingly, ancient bricks generally have a higher porosity compared with ordinary rocks owing to the abundance of pores in the brick [13], which benefits the brick's fluid absorption ability, including water absorption and saline solution absorption. Moreover, the slight

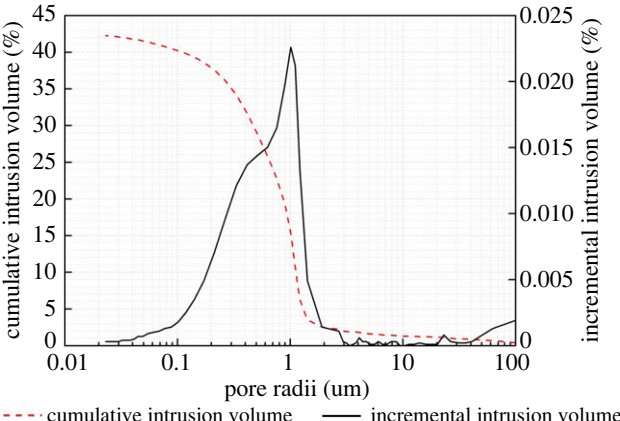

**Figure 3.** Demonstration of the PSD curve of compact sample derived from a Ping Yao ancient brick.

**Table 4.** Mercury (Hg)-porosimetry of five compact samples derived from Ping Yao ancient bricks.

| pore radii (µm) | intrusion volume (ml g⁻¹) | | | | |
| --- | --- | --- | --- | --- | --- |
| | Hg-1 | Hg-2 | Hg-3 | Hg-4 | Hg-5 |
| >5 | 0.0059 | 0.0149 | 0.006 | 0.0077 | 0.0088 |
| 2.5–5 | 0.0028 | 0.0162 | 0.0055 | 0.003 | 0.0017 |
| 1–2.5 | 0.0170 | 0.1389 | 0.0588 | 0.0633 | 0.0067 |
| 0.5–1 | 0.0258 | 0.044 | 0.0682 | 0.0656 | 0.0588 |
| 0.1–0.5 | 0.0471 | 0.0568 | 0.0836 | 0.055 | 0.0909 |
| 0.05–0.1 | 0.0029 | 0.0033 | 0.0084 | 0.0056 | 0.0134 |
| <0.05 | 0.0022 | 0.0011 | 0.0046 | 0.0044 | 0.0067 |
| total | 0.1037 | 0.2752 | 0.2351 | 0.2046 | 0.187 |

difference between the open porosity ($p_o$) and total porosity ($p$) indicates the good pore connectivity of the ancient brick that would benefit the capillary action of liquids.

The ancient brick has a relatively high level of both free water absorption ($\omega_a = 18.40\%$) and forced water absorption ($\omega_s = 23.55\%$). This result is consistent with the small difference between the open porosity and total porosity. Moreover, the saturation coefficient of the ancient brick ($C_s = 0.78$) is smaller than the critical value of 0.9, suggesting that the ancient brick has good frost resistance [12,13].

### 5.2.2. Pore size distribution

The PSD is a principal factor that controls the uptake and transport of liquid within a rock or brick. This factor is closely related to the frost resistance and salt crystallization of masonry materials [21,28–31].

The PSD curves are unimodal and concentrated in pores with radii between 0.1 and 5 µm, especially in the range of 0.1–2.5 µm. As shown in table 4, pores less than 0.5 µm, 0.5–5 µm and greater than 5 µm accounted for 22.24–59.36%, 35.94–72.35% and 2.55–5.69% of the total connected porosity, respectively. Figure 3 demonstrates the PSD curve of a compact sample. The authors also measured open porosity and PSD of the mortar that filled two bricks in the wall. The porosity and PSD results of bricks suggest a greater permeability of bricks compared with mortar (lime or soil–lime mixture), because the mortar has smaller open porosity (14.69%) and the majority of pores are also smaller. In the light of the explanation in the ICOMOS-ISCS [8], the erosion of the brick–mortar interface in figure 2 is also probably related to the alveolization (or coving) caused by different permeability between mortar and brick.

The presence of large volumes of pores smaller than 0.5 µm is detrimental in both freeze–thaw and salt crystallization processes [32,33]. It is well accepted that micropores facilitate solution suction and the generation of high crystallization pressures [34,35] owing to the high saturation coefficient (close to 1) and insufficient space for crystal expansion. Masonry materials with abundant large pores (0.5–5 µm)

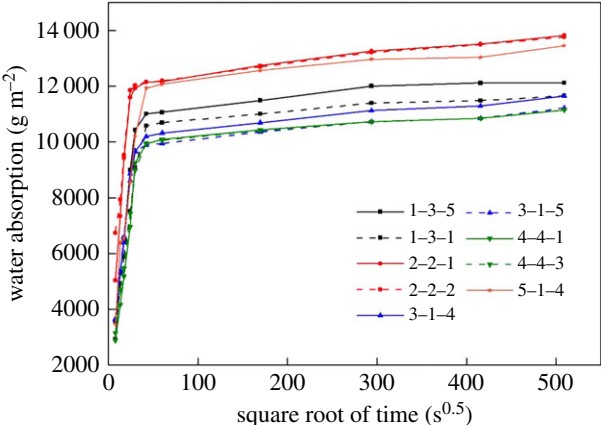

**Figure 4.** Water absorption coefficient by capillarity of nine compact samples derived from Ping Yao ancient bricks.

are generally more resistant to freeze–thaw cycles and salt crystallization (excluding those cases in which the material has a bimodal PSD, which also contains very small pores), as those pores have weak suction and adequate space for the release of crystallization pressure [32]. The pores with radii larger than 5 µm are normally not filled with salt crystals. Yu & Oguchi [21] proposed the salt susceptibility index (SSI) regarding the PSD and classified the resistance of masonry materials into six categories: exceptionally salt resistant (less than 1), very salt resistant (1–2), salt resistant (2–4), salt prone (4–10), very salt prone (10–15) and exceptionally salt prone (15–20). The SSI is defined as:

$$SSI = (I_{Pc} + I_{Pm0.1}) \times (Pm5/P_c),$$

where $I_{Pc}$ is the index of the connected porosity, $I_{Pm0.1}$ is the index of the porosity of pores less than 0.1 µm in radius, $P_c$ is the total connected porosity and $Pm5$ is the porosity of pores less than 5 µm in radius.

The SSIs of samples Hg-1 to Hg-5 are calculated to be 5.66, 10.40, 10.72, 10.59 and 9.53, corresponding to the classes of salt prone, very salt prone, very salt prone, very salt prone and salt prone, respectively. Judging from the PSD and SSI results, the brick samples from the ancient city of Ping Yao are classified as susceptible to salt crystallization.

### 5.2.3. Coefficient of water absorption by capillarity

Figure 4 shows the coefficient of water absorption by capillarity in nine brick samples. The mean value of the samples is $285.7 \pm 17.4 \, \text{g m}^{-2} \, \text{s}^{-0.5}$. One particularly noteworthy detail is, the ancient brick has a greater coefficient of water absorption by capillarity compared with ordinary or even porous rocks.

This finding indicates that the ancient brick has a high speed of water absorption and that the interconnected capillaries inside the brick are beneficial to the rise of damp [36]. This explains the phenomenon of the capillary height of ancient walls being as high as 3 m in some cases. Interestingly, the water absorption coefficients of different brick samples show variation in both rising speed and rising height, which may explain the difference in the rising height observed in different walls. The different speeds and heights in capillary rise may be a result of non-homogeneous compositions of bricks [37].

### 5.2.4. Uniaxial compressive strength

The UCS values of the Ping Yao ancient brick are shown in table 5. Differences of UCS values among samples from the same brick range from 3.03% (brick 5–1) to 24.60% (brick 1–3). Differences of UCS values among samples from different locations can be significant, e.g. sample 2–2–7 is only 5.4 MPa while sample 4–5–5 is as high as 28.7 MPa. Compared with samples from locations 1, 2 and 5, the samples from locations 3 and 4 have higher UCS values, indicating better load-bearing capacity for these bricks. The phenomenon is believed to be caused by different firing temperatures, which will be presented in the following section.

## 5.3. Firing temperature

The relationship between the magnetic susceptibility and re-firing temperature is shown in figure 5a. The squared first derivative of the magnetic susceptibility (figure 5b) is plotted against the stepwise

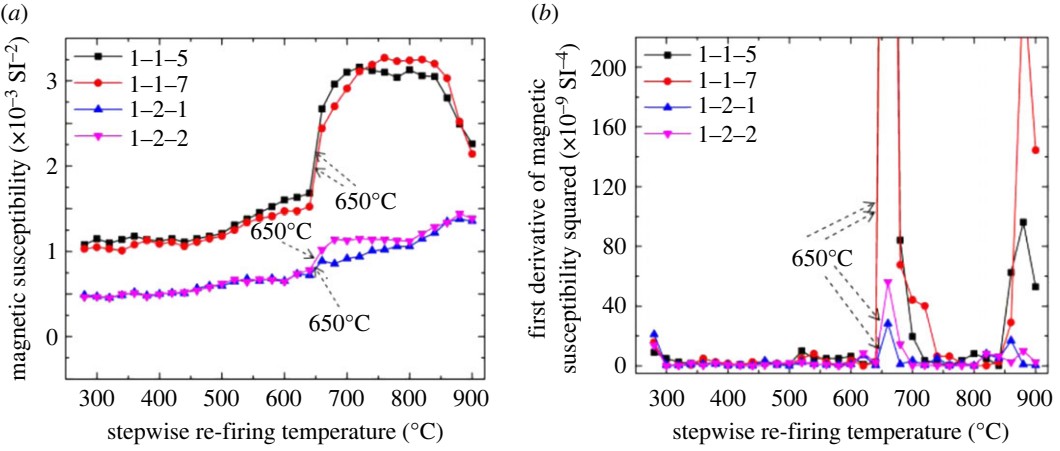

**Figure 5.** (*a,b*) Demonstration of plots used in the experimental determination of the maximum firing temperatures of Ping Yao ancient bricks.

**Table 5.** Uniaxial compressive strength and maximum firing temperatures of brick samples derived from five locations in the ancient city of Ping Yao.

| sampling location | sample ID | UCS (MPa) | sample ID | temperature (°C) |
|---|---|---|---|---|
| 1 | 1–3–5 | 9.5 | 1–1–5, 1–1–7 | 650 |
| | 1–3–6 | 12.6 | 1–2–1, 1–2–2 | 650 |
| 2 | 2–2–3 | 6.1 | 2–1–3, 2–1–4 | 670 |
| | 2–2–7 | 5.4 | 2–2–7, 2–2–8 | 670 |
| 3 | 3–1–1 | 13.5 | 3–3–1, 3–3–5 | 850 |
| | 3–1–5 | 12.2 | 3–4–6, 3–4–8 | 850 |
| 4 | 4–5–1 | 24.5 | 4–1–3, 4–1–6 | 870 |
| | 4–5–5 | 28.7 | 4–4–2, 4–4–6 | 870 |
| 5 | 5–1–2 | 12.8 | 5–1–5, 5–1–6 | 690 |
| | 5–1–3 | 13.2 | 5–2–4, 5–2–6 | 690 |

re-firing temperature to amplify the change of magnetic susceptibility, and therefore to determine the maximum firing temperatures more accurately. The maximum firing temperatures of the tested ancient blue bricks are identified as falling within either the range of 650–690°C or that of 850–870°C (table 5). The relatively low firing temperatures may have led to the poor resistance to salt crystallization and freeze–thaw of Ping Yao brick samples because of high porosity (table 3) and unfavourable PSD (table 4) [5]. Based on the temperature measurements of firing materials in the Ping Yao area using an infrared thermometer (model AR872D, Sigma Technology, Hong Kong, China), the blue bricks tested in this study were probably originally fired by straw or wood in kilns.

## 5.4. Resistance to salt crystallization

Brick samples show poor resistance to salt crystallization in both an $Na_2SO_4$ solution and an $MgSO_4$ solution. The samples are all damaged after four to five cycles. The weight evolution of samples in the three solutions is presented in figure 6. The increase in mass before damage is caused by salt crystals. The damage modes of bricks (figure 7) in these solutions are mainly surface scaling and powdering, which are similar to the damage modes observed in the ancient city of Ping Yao during the field investigation.

The brick samples share similar weathering patterns during the solution treatment. The weathering starts with surface bleaching and an increase in mass as a result of salt absorption in the brick samples. Subsequently, more severe weathering phenomena appear, including rounded corners, crack

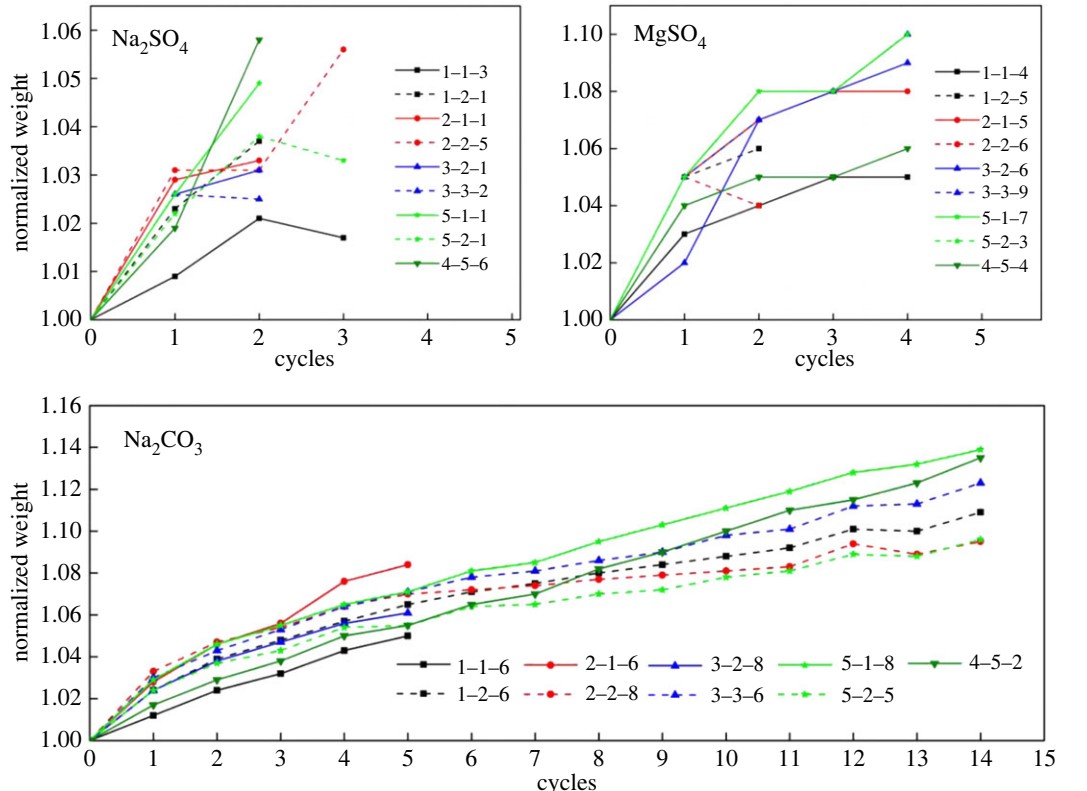

**Figure 6.** Weight evolution of the compact samples in the dry state when immersed in 14% $Na_2SO_4$, $MgSO_4$, and $Na_2CO_3$ solutions. For each sample, the normalized weight is calculated as the weight of the sample after cycling divided by the original weight of the sample. The last dot of each line denotes the normalized weight and corresponding cycles of the sample before failure.

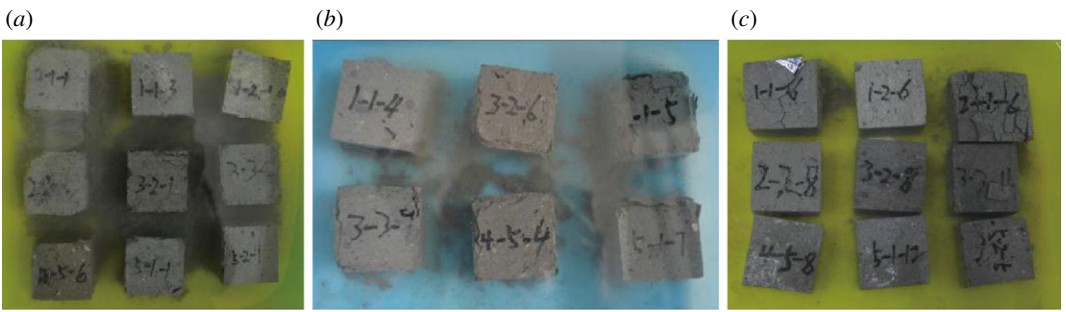

**Figure 7.** Failure modes of brick samples in the (*a*) $Na_2SO_4$, (*b*) $MgSO_4$, and (*c*) $Na_2CO_3$ solutions.

development and surface peeling, followed by angular destruction, crack interconnection, surface scaling and sample partial disintegration.

## 5.5. Frost resistance

Figure 8 shows the evolution of the dry weight, free water absorption and forced water absorption of compact samples in a dry state after a freeze–thaw treatment. The results reveal that freeze–thaw cycles can lead to an increase in pore space and/or connectivity, resulting in an increase in the water absorption of the samples [38]. Interestingly, freeze–thaw cycles also cause increases in the saturation coefficient of 0.781, 0.899 and 0.906 at the 0th, 5th and 10th cycles, respectively. This result indicates an increase in micropores. The influence of the treatment on weight loss is negligible (less than 0.5%).

Unlike damage from salt crystallization (figure 7), the brick samples present a failure mode of macrocracking in the freeze–thaw cycles (figure 9). After the 7th cycle, one sample presents transverse cracks, and another two samples present cracking in corners. After the 10th cycle, three more samples present traverse cracks.

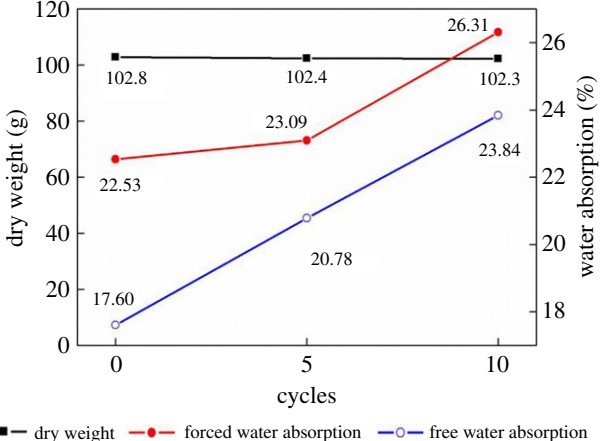

**Figure 8.** Evolution of the average weight and average water absorption of nine compact samples in the dry state after the 0th, 5th and 10th freeze–thaw cycle, respectively.

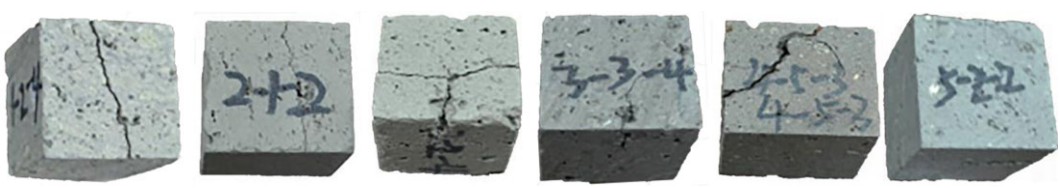

**Figure 9.** Photographs of six damaged samples after 10 freeze–thaw cycles. The failure modes caused by freeze–thaw cycles are mainly macrocracking.

# 6. Discussion on weathering mechanisms and anti-weathering strategies

## 6.1. Effect of salt crystallization

The weathering types and the basic properties of bricks (e.g. porosity, water absorption, PSD and SSI, and salt crystallization test) indicate that salt crystallization is an important contributor to brick damage [39,40]. A schematic of the weathering process of the brick walls in the ancient city of Ping Yao is shown in figure 10. Detailed mechanisms are presented as follows [41]:

 (i) the damp rises along interconnected pore networks in ancient brick walls from the foundation soil until the upward capillary force and downward gravity achieve balance. The soluble salts are also mobilizing upwards together with the rising damp in this process. The pores mainly have radii less than 5 µm (especially 0.1–1 µm). The damp can reach a height up to several metres;

 (ii) under the effect of wind and sunlight, the moisture in the wall evaporates, leaving soluble salt stranded in the surface area of the brick;

 (iii) moreover, in dry seasons, owing to a moisture reduction and/or a temperature decrease, the soluble salt begins to absorb water vapour and form salt crystals, such as hexahydrite and epsomite. The volume increase during salt crystallization (e.g. subfluorescence) often applies tension in the brick pores and thus causes pore disruption and substrate damage; and

 (iv) the repeated change in environmental factors around the walls (e.g. the variation of humidity and the alteration of negative and positive temperature between summer and winter) leads to increasing amounts of salt left on the wall surfaces as well as in the brick pores, causing the tension in the brick pores to continue increasing.

According to Steiger [31], even a relatively minor degree of supersaturation can produce crystallization pressures that induce tensile stresses exceeding values of 10 MPa. The growth in pressure owing to the process of salt crystallization leads to the scaling and crumbling of the bricks.

 The soluble salts enriched in the walls of the ancient city of Ping Yao have four main sources: foundation soil and underground water; filling and cementing materials of walls (e.g. lime, clay and water); raw clay used in brick production; and atmospheric precipitation and anthropogenic air pollutants.

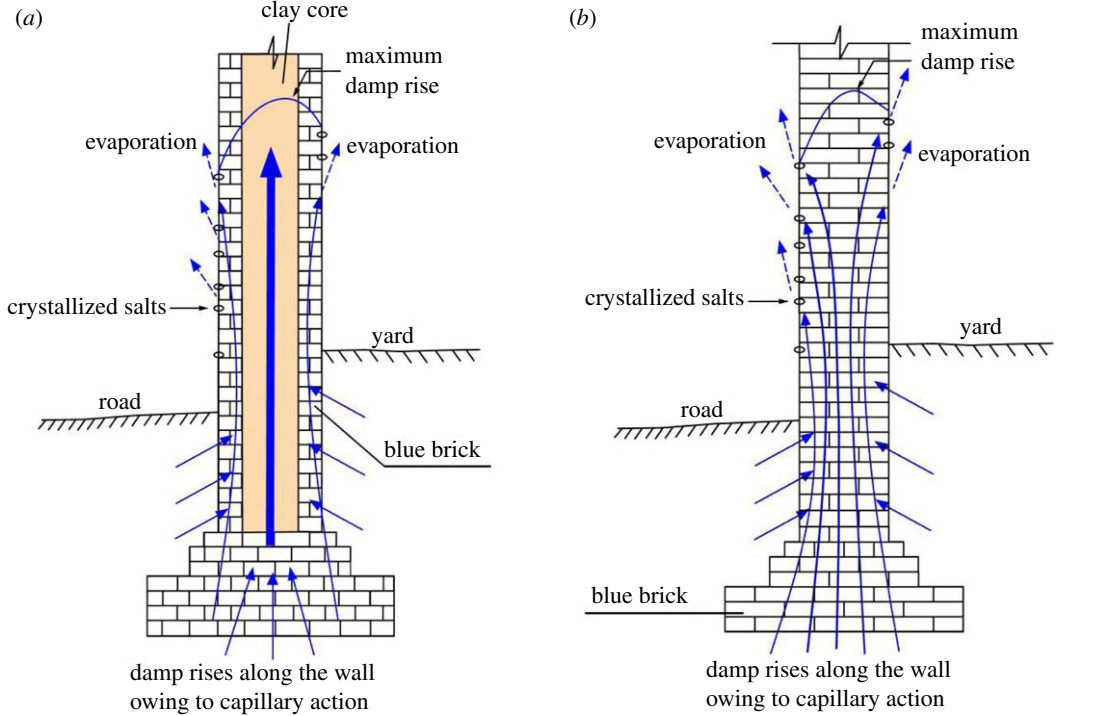

**Figure 10.** Schematic of the rising damp and salt crystallization in two types of ancient brick walls in the ancient city of Ping Yao. The walls containing clay cores (*a*) allow for easier transport of damp, and the zone of rising damp will often be higher compared with those of walls without clay cores (*b*).

The damp in the Ping Yao ancient walls comes mainly from capillary rise, rather than rainfall. This argument can be verified by the fact that the height of the capillary rise in the walls is not immediately associated with the rainfall in the study area. For instance, a drizzle and moderate rain lasted for one week from 17 June to 23 June 2013, in the Ping Yao area, yet no obvious damp increase in the walls was observed, as measured by the moisture meter.

## 6.2. Effect of freeze–thaw cycles

The brick samples were all damaged after 10 freeze–thaw cycles, which denotes that the freeze–thaw cycles is another contributor to the weathering of Ping Yao ancient bricks. Frost damage could possibly be a combination of three principal theories [42]: the volumetric expansion (approx. 9%) of water on freezing, the crystallization or 'ice lens theory', and the hydraulic pressure theory. Even though the laboratory results indicate that the freeze–thaw cycles are detrimental to the preservation of the ancient brick, several inconsistencies still remain in this pioneer study of ancient brick weathering, as discussed below:

(i) to assess the seasonal influence of damp on the brick walls, an IMKO HD2-Mobile Moisture Meter equipped with TRIME-PICO probes was used to measure the moisture content (MC) of the joint mortars between bricks. The MC tests were carried out at four different locations on 20 June (in the rainy season) 2013. In each location, 4–10 different spots ranging from 0.5 to 1.5 m above ground level were tested. The tests were repeated in the same locations on 26 October (in the dry season) 2013. On 20 June, the maximum MCs (%) of the four measurement locations were 28.49, 65.94, 27.82 and 35.94, respectively. On 26 October, the corresponding data were reduced to 6.10, 15.30, 7.52 and 10.89, respectively. Although the rainfall/snowfall amount is small in winter, there are rainfall periods that overlap freezing temperatures (i.e. October, November and March) in cold seasons. The existence of a small amount of water in walls may still cause damage to bricks owing to the brick's PSD profile that intensifies the effects of freeze–thaw; and

(ii) on the other hand, the laboratory experiments show significant damage caused by freeze–thaw cycles, but the frost heaving damage modes in the experiment (figure 9) are not quite consistent with the modes observed on-site. In the laboratory, the typical frost heaving damage mode of

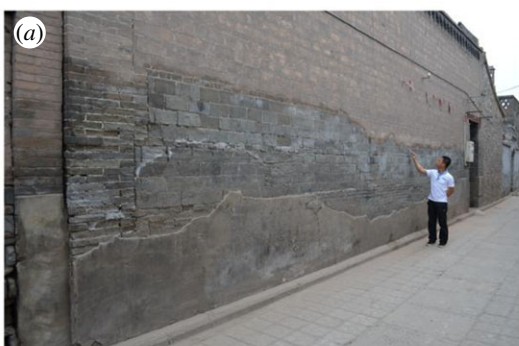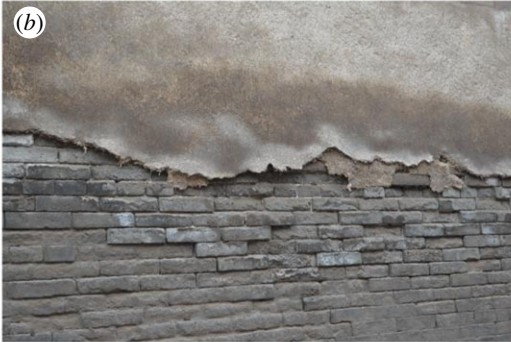

**Figure 11.** The practice of (*a*) smearing cement or (*b*) clay and plant fibres on the ancient brick in walls failed to prevent weathering of the ancient bricks.

the ancient bricks is macroscopic cracking, and the cleavages of different samples are at roughly equal levels (figure 9). The failure modes observed on-site include contour scaling and granular disintegration, which are more similar to the failure modes caused by salt crystallization.

Therefore, it can be confirmed that freeze–thaw is a factor accounting for the weathering of ancient blue brick in Ping Yao. However, according to present information, whether the contribution of freeze–thaw cycles outweigh that of salt crystallization to the weathering is still unclear and needs to be further investigated in a next step. To have a quantitative understanding, the MC in brick walls, rainfall condition and underground water condition in wet and dry seasons need to be monitored at more locations; and then laboratory-based freeze–thaw cycles should be conducted on the samples with specific MCs that are based on the field results.

## 6.3. Discussion on anti-weathering strategies

The present protective covering is observed to be made by cement, clay and plant fibres (figure 11) in Ping Yao. However, these covering materials failed to retard the weathering of the brick masonry. In reality, it is unsuitable to cover masonry surfaces with waterproofing materials (e.g. impermeable plasters and bitumen) [43]. The misuse of repair materials may itself damage the cultural relics, and is considered an improper artificial repair.

Field investigation and weathering analyses have proved that damp transport is the major environmental factor threatening the preservation of brick buildings. Damp transport is directly related to the destructive effects of salt crystallization and temperature fluctuations (i.e. freeze–thaw cycles). Therefore, the guidelines of alleviating weathering effects is to decrease the amount and height of rising damp and to reduce the impact of salt crystallization and freeze–thaw. To achieve this objective, methods of blocking the rising damp, desalination and brick replacing should be combined. The properties of blue bricks themselves also should be considered. Studies have confirmed that the firing temperature during brick-making determines the development of the new mineral phases and pore structures in bricks, thus determining their durability [44,45]. To evaluate the influence of firing temperatures on the Ping Yao brick, the authors fired adobes made from Ping Yao clay at 700, 800, 900, 1000 and 1100°C and tested their properties (tables 6 and 7).

In order to reduce the influence of damp, one step is to reduce the moisture supply, another step is to increase the moisture evaporation at the near ground area such that the height within the reach of rising damp can be limited. Research conducted by Guimarães *et al.* [46] and Ahmad & Rahman [47] have proved that ventilation is an effective way to reduce the rising height of damp. Figure 12 presents a possible method to be implemented: (i) the soil near the walls should be replaced by pebbles as pebbles cannot transport damp; and (ii) a water collection pipe should be placed at the lower side of the ditches. This design is expected to serve three purposes: the first is to provide a drainage system and thus reduce the amount of water that infiltrates soils during rainy seasons, the second is to cut off the horizontal moisture transportation path and reduce the moisture source, and the third is to incorporate 'natural ventilation' in the area where the moisture concentrates. Considering that salt crystallization is an important factor in brick weathering, the use of sacrificial bricks as protective layers is suggested. According to the results in tables 6 and 7, it is preferable to select bricks fired at

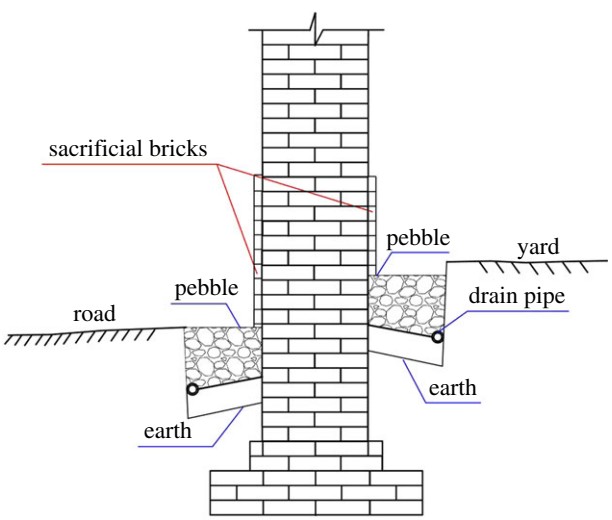

**Figure 12.** Sketch map of damp blocking, ventilation and sacrificial brick layers combined technique in the ancient wall.

**Table 6.** Physico-mechanical properties of bricks made from Ping Yao clay and fired at different temperatures.

| parameters | 700℃ | 800℃ | 900℃ | 1000℃ | 1100℃ |
|---|---|---|---|---|---|
| apparent density ($\rho_b$, g cm$^{-3}$) | 1.79 | 1.77 | 1.77 | 1.76 | 1.94 |
| real density ($\rho_r$, g cm$^{-3}$) | 2.67 | 2.66 | 2.69 | 2.72 | 2.64 |
| open porosity ($p_o$, %) | 32.7 | 33.0 | 33.7 | 33.9 | 25.9 |
| total porosity ($p$, %) | 33.0 | 33.5 | 34.3 | 33.5 | 26.6 |
| free water absorption ($\omega_a$, %) | 15.8 | 16.6 | 16.9 | 17.2 | 9.7 |
| forced water absorption ($\omega_s$, %) | 17.3 | 18.2 | 18.5 | 19.0 | 12.7 |
| saturation coefficient ($C_s$) | 0.912 | 0.913 | 0.916 | 0.907 | 0.764 |
| UCS (MPa) | 13.02 | 14.82 | 19.01 | 12.60 | 23.77 |

**Table 7.** Hg-porosimetry of bricks made from Ping Yao clay and fired under different temperatures.

| pore radii (µm) | percentage of intrusion volume (%) | | | | |
|---|---|---|---|---|---|
| | 700℃ | 800℃ | 900℃ | 1000℃ | 1100℃ |
| >5 | 1.7 | 2.0 | 2.6 | 0.6 | 5.0 |
| 2.5–5 | 0.2 | 0.2 | 0.2 | 0.7 | 0.4 |
| 1–2.5 | 0.3 | 0.2 | 0.0 | 0.3 | 31.8 |
| 0.5–1 | 0.7 | 14.3 | 26.3 | 52.0 | 52.7 |
| 0.1–0.5 | 49.8 | 62.9 | 60.9 | 37.2 | 8.9 |
| 0.05–0.1 | 15.9 | 9.2 | 5.5 | 4.8 | 0.8 |
| <0.05 | 31.4 | 11.1 | 4.5 | 4.4 | 0.5 |
| total | 100.0 | 100.0 | 100.0 | 100.0 | 100.0 |

700℃ as sacrificial substrates to trap higher amounts of salts [5,48] owing to the higher porosity and higher percentages of small pores.

Besides the general methods proposed above, special treatments may be also needed in areas that are severely affected by the weathering phenomena. Desalination methods are suggested to retard the weathering rate of walls colonized by salt crystals. According to previous cases reported in the literature, starch grafted acrylamicle [49] and Westox Cocoon poultice [41] are recommended as the first batch of

desalination materials to conduct field testing with, in order to evaluate the effectiveness and suitability for salt removal in Ping Yao brick walls. For brick replacement, bricks fired at different temperatures should be used as circumstances may require. Based on the results in tables 6 and 7, bricks fired at 1100°C should be used in replacing the load-bearing parts for their low porosity, favourable PSD, low water absorption and saturation coefficient, high strength and pore structures beneficial to salt resistance.

It is also noteworthy that the synergetic anti-weathering methods discussed above are based on the analyses of laboratory experimental results, the methods reported in the literature and the current condition of Ping Yao City observed during site investigation. The effectiveness of the proposed methods and subsequent modifications are yet to be studied on a field-test basis, which is the next step of this study.

# 7. Conclusion

This paper presents the results of the phase one study of our research project. Through on-site investigation and laboratory tests, the weathering types, physico-mechanical properties, and resistance to salt crystallization and freeze–thaw cycles of the bricks in the ancient city of Ping Yao were studied. The weathering factors and anti-weathering strategies of ancient brick have also been discussed. The main achievements are listed as follows:

(i) blue bricks are used as the main building materials in the ancient city of Ping Yao. The parts of the walls within reach of the damp (maximum height greater than 4 m) are severely weathered with the weathering effects of contour scaling, flaking, powdering and salt crystallization;

(ii) the bricks are mainly composed of quartz, feldspar and clay minerals. Small quantities of gypsum and calcite are also found in some bricks. Compared with the concave parts (less durable), the convex parts (more durable) of the brick contain higher amounts of quartz but less clay minerals;

(iii) the bricks have a high water absorption ability with free water absorption and forced water absorption of 18.40% and 23.55%, respectively. The coefficient of water absorption by capillarity is as high as 285.7 $g\,m^{-2}\,s^{-0.5}$. Pores are well developed in the brick, and the pore size is mainly concentrated in the range of 0.1–2.5 µm;

(iv) the salt on the surface of the ancient bricks is mainly composed of hexahydrite, epsomite and sodium nitrate. The durability of the brick is not ideal in terms of its resistance to salt crystallization and frost resistance, as proved by the PSD and SSI results;

(v) the firing temperatures of the ancient brick are identified in the ranges of 650–690°C and of 850–870°C, a factor that accounts for the brick samples' poor resistance to salt crystallization and frost. The ancient bricks were probably originally fired by straw or wood in kilns;

(vi) salt crystallization and the freeze–thaw cycle are two important factors accounting for the brick weathering of Ping Yao ancient bricks. The damp risen from the foundation soil evaporates into the atmosphere under the effect of wind or/and sunlight, leaving soluble salts stranded on the brick surface and forming crystals. The pressure generated in the process can cause brick damage. The freeze–thaw cycles also lead to brick damage owing to overlapping of rainfall/ snowfall and freezing periods. A field monitoring scheme is suggested for further study on the effects of the freeze–thaw; and

(vii) an anti-weathering strategy of 'damp blocking, desalination and brick replacing' is discussed based on the experimental results and site conditions. Specifically, block the damp transport; desalinate the bricks that have already been invaded by salt crystals; for sacrificial layers and parts needing to be replaced, choose replacement bricks fired at suitable temperatures as circumstances may require. The proposed strategy provides a relative narrow range for field testing and long-term monitoring in the next stage.

Data accessibility. Data and materials are included in the electronic supplementary material.
Authors' contributions. Z.J.Z. conceived and designed the study, and carried out the field investigation. J.B.L. carried out the laboratory work and analysed the data. J.B.L. wrote the manuscript. Z.J.Z. critically revised the manuscript. All authors gave final approval for publication and agree to be held accountable for the work performed therein.
Competing interests. The authors declare that there are no competing interests.
Funding. Financial supports came from the National Natural Science Foundation of China (grant no. 41602329) and the Fundamental Research Funds for the Central Universities (grant no. 2652019075).

**Acknowledgements.** The authors would like to thank financial support from the National Natural Science Foundation of China (no. 41602329) and the Fundamental Research Funds for the Central Universities (no. 2652019075). The authors would like to thank Ping Yao Cultural Relics Bureau, Minsu Inn and Yunzhou Inn for providing support in the field investigation and sample collection. The authors would like to thank the American Journal Experts and Miss Claire V. Owens for polishing the language of this manuscript.

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
