## [Reviewer comments · Royal Society Open Science]

Review History

RSOS-191008.R0 (Original submission)

Review form: Reviewer 1

Is the manuscript scientifically sound in its present form?

Yes

Are the interpretations and conclusions justified by the results?

No

Is the language acceptable?

Yes

Do you have any ethical concerns with this paper?

No

Have you any concerns about statistical analyses in this paper?

No

Recommendation?

Accept with minor revision (please list in comments)

Comments to the Author(s)

Generally a good article where the authors have understood that measuring the concentration of salt in the substrate is only one part of the issue, since substrate characteristics determine its resistance to salt decay. However there is less detail on the severity and frequency of environmental fluctuations which are equally important in determining rates of salt damage and this reviewer feels this should be addressed more thoroughly.

Comments on the text:

the term 'folk house' needs definition.

p4(pdf) Ln44+ - useful to know at what temperatures the adobe blocks were fired. In Europe there are both low temp. and high temp. bricks and the process of their manufacture is not as different as claimed with both sharing processes of controlled mixing, moulding, natural drying and firing.

p5(pdf) presents a list of publications on red brick conservation - which is not the subject under discussion and could be made much shorter to help the point made in the next paragraph on p6 (pdf) that there are few studies on blue bricks. Where the mineral compositions are found to be similar this should all be detailed in the relevant section on p13 (pdf).

p6 (pdf) Ln 39+ there is a bold and ultimately incorrect claim about 'safeguarding' blue bricks that needs qualification (see later).

the 'average' annual figures given on p7(pdf) are not useful and its the min/max ranges and length of exposure that are important. For example year long microclimate monitoring would show if there were particularly long and deleterious wetting-drying and freeze-thaw cycles etc esp. as on p17 (pdf) you say that

"The repeated change in environmental factors around the walls (e.g., humidity and temperature) leads to increasing amounts of salt left on the wall surfaces as well as in the brick pores, thus the tension in the brick pores continues to increase"
but do not state where this information on rapid environmental change comes from.

Given the 'averages' given before, the description of weathering types is extremely brief and conflated with rising damp. This section needs a little more explanation relating to particular mechanisms observed in the locale. It incorrectly lists the effects of weathering - contour scaling, flaking etc - as 'weathering types'.

The analyses is where you exhibit greatest confidence and the results confirm standard interpretation eg. greater water transport when smaller capillary radii and that distribution is never regular. Within the given ranges of 0.1 to 5 μ m pore radii its interesting to note if you can confirm Winkler's(1994) findings that capillaries $\leq 0.1 \mu$ m are generally unable to absorb water (so the greater % of their distribution, the lower the SSI)?

p14 (pdf) what you haven't reported on is the type of mortar used between the bricks - this is essential to understand as if the mortar is more impermeable than the bricks then the concave decay described here might be caused by the harder mortar (it looks like a mastic in Fig 2) rather than just the differences in manufacturing mixes to which you attribute it. I think this omission needs looking at before publication - cf. definition of 'coving' as an erosion feature consisting in a single alveole developing from the edge of the stone block often caused by harder/more impermeable pointing on p28 of

https://www.icomos.org/publications/monuments_and_sites/15/pdf/Monuments_and_Sites_15_ISCS_Glossary_Stone.pdf

re. p16(pdf) 6.2.3 - the speed and height variation in capillary rise are to be expected as the bricks are non-homogeneous in composition - Camuffo (2014) has some interesting work in this area.

re. p17(pdf) re. TRM/archaeomagnetic dating - were analyses made to identify any local spikes or rapid changes in geomagnetic activity etc in the 14-17th C period? This needs consideration given the claim that 'the original maximum firing temperatures of the brick samples are identified' in your covering letter.

re. 7.1 - this needs to take account that rising damp causes the mobilisation of soluble salts in the bricks leading to their crystallization. Crystallization should be differentiated more clearly between efflorescence and subfluorescence - its the latter, as you identify, that causes pore disruption and substrate damage.

p18 (pdf) Bear in mind that if the front of the dampness edge is not generally horizontal the cause of the wetting could also be from elsewhere - although Fig 11 shows its all horizontal on that elevation.

p19 - I did think the interpretation of one set of samples in the rainy and the dry seasons was methodologically insufficient to conclude that free-thaw cycles were not significant re. brick damage - this should be recharacterised as needing further study/research, especially as here you are quantifying your results against rainfall which you have already said is not the source of the dampness - you need to redo this with reference to groundwater conditions and rising damp. Furthermore, any experimental results are never isomorphic with action/damage in the field, so I am not convinced by your comparison made in 7.2 b) especially as flaking etc can be caused by freeze-thaw action.

on p22(pdf) you make the correct claim about not using waterproofing agents but then go on to say that "it is highly unsuitable to cover masonry surfaces with lime" and reference one article on stress histories in sandstone as created by various things including lime renders - however many building conservation treatments include the use permeable lime washes and renders as sacrificial coatings especially as they are seen to help salts crystallize out in them and not the stone substrate, which is something you do not address. Of course, whether you would want to cover historic brickwork in lime is another matter.

p22 (pdf) The "anti-weathering strategy" is the weakest part of the article. You opt for the convention of installing French drains to mitigate the rising damp but then suggest removing rows of bricks and extant lime/clay mortar beds/pointing as shown in Fig 12 - historic in themselves - perhaps using the 'total cut' method, and replacing them with "waterproof materials" such as "steel [48], lead plates, bitumen-based membranes, polyethylene [49], and polyester-based membranes." This is dramatic stuff, however your ref. re. steel is slightly incorrect as the authors cited concluded that: "The groundwater level should be checked; if the level is likely to be higher than where the blocking of capillary suction is planned, the blocking should be watertight, e.g. not steel plates". This proviso applies here to all the materials suggested and needs to be built into any engineering works as proposed (but whihc you do not assess).

Of course what you propose is damaging to the cultural heritage you are trying to protect - as Doehne et al. note - ie. wire sawing out double rows of bricks and mortar at 2 different heights is very invasive as well as expensive and not without risk.

However, even though you suggest replacing the jointing with impermeable materials and old bricks with new more dense impermeable bricks fired at 1100 degrees C (p23 pdf) you also suggest the use of sacrificial bricks which I think sits oddly with the idea of the impermeable double layer, no? Why is this necessary?

So I'm not convinced you have thought this through quite enough from an engineering or conservation perspective and think these suggestions need revisiting and given more thought and properly phased in experimental regimes.

Furthermore you should be at least quantifying whether the extant pointing is impermeable or not - and if so first setting up a test area and replacing it with softer lime mortar to observe if it mitigate the effects of any decay.

You should also be proposing to set up other test areas to trial French drains, both with and without colour-matched sacrificial lime coats and with and without replacing the extant pointing with softer lime mortars ie. all standard practices. Only after these have failed to mitigate the effects of the rising damp should tests then look to more dramatic interventions, but also made first in controlled test sections.

Obviously desalination as proposed in the article to take place once your impermeable barrier has been installed but it had to be thorough with poultice materials well matched to the substrate and salts being treated so this also needs more assessment than apparent in the article.

All of this is well within the scope of a section on further research and I suggest this as a way out of otherwise appearing to have rushed into unsubstantiated claims (p6 pdf) about having successfully 'safeguarding' the blue bricks.

Review form: Reviewer 2

Is the manuscript scientifically sound in its present form?

No

Are the interpretations and conclusions justified by the results?

No

Is the language acceptable?

Yes

Do you have any ethical concerns with this paper?

No

Have you any concerns about statistical analyses in this paper?

Yes

Recommendation?

Reject

Comments to the Author(s)

Overall this is an interesting study, addressing a relatively little-researched brick type. The authors have gathered a lot of data across various methods. However, there are issues with this manuscript, in particular in the methodology and associated conclusions, which make it unsuitable for publication in its current state.

The authors mention three different sample types, but throughout the manuscript it is not made clear which samples are analysed. While the reader can work it out with some effort, it would be easier if the authors provided a clear overview of which samples was used for which test and how these findings are tied together. As it stands the study presented here is rather fragmented. This therefore undermines the credibility of the conservation suggestions made by the authors. The methodology is not always clearly explained and justified, which again makes it more difficult to follow the narrative laid out by the authors. In fact, some methodologies, such as the repeated heating of blocks, are not justified at all which undermine the interpretations set out in

the discussions. Towards the end of the manuscript there are quite a lot of frustrating statements, especially when the authors introduce environmental conditions without providing any sound environmental assessments.

P4

The authors need to embed further literature in their introduction. The second paragraph in particular, which describes the firing process for blue bricks, does not contain any further references, and lacks in detail such as firing temperature etc. There does seem to be considerable overlap between traditional red clay brick production (as seen widely around Europe) and blue brick production, yet the authors do not clearly differentiate this. Why not use more of the available literature on red bricks?

P5

L7 – 18 this paragraph is very anecdotal – give more examples and theoretical grounding.

L21- P6L14 There are two issues with this paragraph: [1] the examples cited are predominantly on European red brick, yet the authors have already stated that the blue brick is different – this has not been worked into the narrative presented here and [2] the studies are cited but there is no summary of their findings. So it is not clear how useful these studies are in providing the theoretical framework for the study presented here.

P6

Where are the samples in Fig 2 located in Fig 1?

L18 – 47 Again, there is little citation of the wider literature. It is also unclear what the authors mean with ‘this study not only safeguards brick masonry...’

P7

L15: This needs a reference

L31: Define ‘great preservational value’ – are the authors talking about the value placed on the heritage by the local community?

L59: What evidence do the authors have for the height of the wetting zone?

P8

L5-7: This is a rather vague statement; do these classifications refer directly to the blue brick or is this simply the general list provided by ICOMOS?

L25: ‘The samples belong to two sub-types according to the sampling parts:’ – it is unclear what the authors mean by this

L21-33: The sampling of the bricks needs to be explained in much more detail. It is unclear where the samples were taken, how they were selected and how representative they are. The authors have already flagged up the importance of rising damp, yet do not provide detail on the location of the samples within the wider wall.

L35 – 38: How were these samples selected and obtained?

L41 – 55: The origins and potential weathering history of these bricks seem dubious. Do the authors have further information about their production history and origins?

P9:

L13: Why are they indispensable? Make sure that statements like these are properly explained.

L16: Which samples are the authors referring to? Sample set 1 or 3?

P10:

L12 – 18: Why do the authors not take the elasticity of the material into account when considering the potential damage of frost action?

L 24 – 32: Sample size and drying methods are not justified.

L44: What is ‘pure water’? Distilled or deionised water?

L39 – 57: Why did the authors change the sample size from the one used for the mercury intrusion test? What were the time intervals and how were they determined?

P11:

L8: The use of the word ‘inevitable’ is very odd – the authors really need to be far more thorough in justifying all methods.

L13: Why 0.5 MPa/s? What does this load represent?

L18: This is a very small sample size compared to the original size of the bricks. How do the authors translate these findings to the behaviour of the bricks in situ?

L25 – 60: This whole section is poorly justified. There seems to be no correlation between the cyclical / repeated firing and the production of the original bricks. The authors acknowledge that firing leads to changes in the brick's mineral phases, yet this is not taken into account during repeated incremental firing. This repeated firing could have altered the bricks sufficiently to influence any further testing.

P12

L25: Why these salts? Are these the ones found in situ?

L28-41: Again, the methods are poorly justified here – why the 2 hour cooling phase? Why not 1 hour or 12 hours? What salts were used in the solution? (this is not clear from the preceding paragraph). There is no indication of use of control samples.

L46 – 58: What is 'pure water'? Which samples were used? What would the implications of their potential weathering history be?

P13:

L 31-33: 'Gypsum is the product...atmospheres' It is unclear what the function of this sentence is here.

L39-47: The heterogeneity of the bricks stated here undermines the validity of the general conclusions drawn until now, and the general conclusions of the study. The authors need to address this issue clearly.

P14

L7: The heterogeneity of the bricks is not translated in the results presented in table 3.

L14: What are 'ordinary rocks'? L14 – 24 Make little sense to the reader.

L27 – 38: The authors have switched back to general observations, yet do not address the variability of the bricks!

L56-59 / Table 4: The results presented in the text and in the table are not correlated well – it is unclear how the (large!) % ranges presented in the text relate to the results presented in the table.

P15

L4: Which sample?

L20-24: The authors report that the bricks contain anywhere between 22-59% small pores in total connected porosity, which would fall well within the bimodal pore size distribution category. They therefore exclude their own study samples here!

L54: The classifications make little sense when presented here in the text.

P16

L10: Again, this statement ignores the heterogeneity of the bricks

L14: What constitutes an 'ordinary' or 'even porous' rock?

L19: Rising damp is strongly connected with temperatures, and thus evaporation rates. It is not just a function of the capillarity of the bricks. The authors need to take into account the environmental conditions of the sample sites, and also provide actual measurements to justify their claims of capillary rise heights.

L36 – 52: The authors give the impression here that their samples are drawn from a varied pool of bricks, with large variability in the original material as well as production method. It is therefore difficult to justify their interpretations considering the limited sampling they have undertaken and the variability in the nature of the samples.

P17

L4-27: See my previous point regarding cyclical and incremental testing vs the actual firing method used in the production of these bricks.

L36: What is classified as 'damage'? Small cracks or structural failure? Pitting of the surface? The graphs do not clearly indicate when the sample is damaged – is it when the line stops?

L48 – 58: Can the authors relate the variability in porosity to the response of the block to the salt weathering?

P18:

L14: In all samples? Are the results pooled? This again ignores the variability in the samples.

L24 – 31: Which samples? The authors are very vague in their statements here

L43-48: This is not clear from the presented results – the freeze thaw tests indicate macro-cracking. The authors do not provide detailed observations of salt weathering across the study site, nor do they provide sufficient information on diurnal temperature fluctuations throughout the year to discard freeze thaw weathering.

L58 – 60: This is the first time the authors introduce clay cores in the walls, which changes the potential behaviour of the bricks.

P19 and P20

This is the first time the authors take environmental conditions into account, and provide little in the way of measurements and observations of justify their statements. The study should be rewritten and restructured with environmental conditions taken into account for the authors to justify making such statements.

P20

L18 – 43: Freeze thaw damage effectiveness is dependent on repeated above and below zero temperatures, more so than moisture content. The authors have not measured temperatures. Therefore, their statements regarding the importance of freeze thaw are not justified.

L46: Frost heave is not the same as frost-induced cracking. Please consult the literature on ‘frost heave’

P21

L3-20: The authors invalidate their own findings, and admit the issues that I have already flagged up in this review. So, how valid are the conclusions drawn from this study? In my view they are not justified.

In light of the problems flagged up regarding the variability of the material in the test results and the discrepancy between in situ conditions, environmental conditions and the laboratory samples it is very questionable whether or not the ‘anti-weathering strategies’ are justified. In my view they are not.

Decision letter (RSOS-191008.R0)

16-Jul-2019

Dear Dr zhang:

Manuscript ID RSOS-191008 entitled "Characteristics, weathering mechanisms and anti-weathering strategies of the traditional Chinese blue brick from the Ancient City of Ping Yao" which you submitted to Royal Society Open Science, has been reviewed. The comments from reviewers are included at the bottom of this letter.

In view of the criticisms of the reviewers, the manuscript has been rejected in its current form. However, a new manuscript may be submitted which takes into consideration these comments.

Please note that resubmitting your manuscript does not guarantee eventual acceptance, and that your resubmission will be subject to peer review before a decision is made.

Your resubmitted manuscript should be submitted by 13-Jan-2020. If you are unable to submit by this date please contact the Editorial Office.

Kind regards,

Andrew Dunn
Royal Society Open Science Editorial Office
Royal Society Open Science
openscience@royalsociety.org

on behalf of Prof R. Kerry Rowe (Subject Editor)
openscience@royalsociety.org

Associate Editor Comments to Author:

Thank you for this interesting piece. We've received two substantial reviews on your work, and while each reviewer clearly sees some merit in your work, there are major concerns at present regarding, for instance, the methodology you've employed. We cannot accept the paper for further consideration in its current form, but we would be willing to consider a resubmission on condition that you not only take onboard the feedback from the reviewers in a revised manuscript but also provide a thorough scientific response to each of the queries and critiques the reviewers provide. Given the extent of the changes necessary, it is likely the next iteration of your paper will be substantially different from the original submission, hence the Editors recommending a reject/resubmit, rather than a revision (which would not provide you sufficient time to reasonably modify your paper). If we do not hear from you within the deadline, we will assume you've withdrawn the paper, otherwise good luck - we'll look forward to receiving the resubmission.

Reviewers' Comments to Author:

Reviewer: 1

Comments to the Author(s)

Generally a good article where the authors have understood that measuring the concentration of salt in the substrate is only one part of the issue, since substrate characteristics determine its resistance to salt decay. However there is less detail on the severity and frequency of environmental fluctuations which are equally important in determining rates of salt damage and this reviewer feels this should be addressed more thoroughly.

Comments on the text:

the term 'folk house' needs definition.

p4(pdf) Ln44+ - useful to know at what temperatures the adobe blocks were fired. In Europe there are both low temp. and high temp. bricks and the process of their manufacture is not as different as claimed with both sharing processes of controlled mixing, moulding, natural drying and firing.

p5(pdf) presents a list of publications on red brick conservation - which is not the subject under discussion and could be made much shorter to help the point made in the next paragraph on p6 (pdf) that there are few studies on blue bricks. Where the mineral compositions are found to be similar this should all be detailed in the relevant section on p13 (pdf).

p6 (pdf) Ln 39+ there is a bold and ultimately incorrect claim about 'safeguarding' blue bricks that needs qualification (see later).

the 'average' annual figures given on p7(pdf) are not useful and its the min/max ranges and length of exposure that are important. For example year long microclimate monitoring would show if there were particularly long and deleterious wetting-drying and freeze-thaw cycles etc esp. as on p17 (pdf) you say that

"The repeated change in environmental factors around the walls (e.g., humidity and temperature) leads to increasing amounts of salt left on the wall surfaces as well as in the brick pores, thus the tension in the brick pores continues to increase"
but do not state where this information on rapid environmental change comes from.

Given the 'averages' given before, the description of weathering types is extremely brief and conflated with rising damp. This section needs a little more explanation relating to particular mechanisms observed in the locale. It incorrectly lists the effects of weathering - contour scaling, flaking etc - as 'weathering types'.

The analyses is where you exhibit greatest confidence and the results confirm standard interpretation eg. greater water transport when smaller capillary radii and that distribution is never regular. Within the given ranges of 0.1 to 5 μ m pore radii its interesting to note if you can confirm Winkler's(1994) findings that capillaries $\leq 0.1 \mu$ m are generally unable to absorb water (so the greater % of their distribution, the lower the SSI)?

p14 (pdf) what you haven't reported on is the type of mortar used between the bricks - this is essential to understand as if the mortar is more impermeable than the bricks then the concave decay described here might be caused by the harder mortar (it looks like a mastic in Fig 2) rather than just the differences in manufacturing mixes to which you attribute it. I think this omission needs looking at before publication - cf. definition of 'coving' as an erosion feature consisting in a single alveole developing from the edge of the stone block often caused by harder/more impermeable pointing on p28 of https://www.icomos.org/publications/monuments_and_sites/15/pdf/Monuments_and_Sites_15_ISCS_Glossary_Stone.pdf

re. p16(pdf) 6.2.3 - the speed and height variation in capillary rise are to be expected as the bricks are non-homogeneous in composition - Camuffo (2014) has some interesting work in this area.

re. p17(pdf) re. TRM/archaeomagnetic dating - were analyses made to identify any local spikes or rapid changes in geomagnetic activity etc in the 14-17th C period? This needs consideration given the claim that 'the original maximum firing temperatures of the brick samples are identified' in your covering letter.

re. 7.1 - this needs to take account that rising damp causes the mobilisation of soluble salts in the bricks leading to their crystallization. Crystallization should be differentiated more clearly between efflorescence and subfluorescence - its the latter, as you identify, that causes pore disruption and substrate damage.

p18 (pdf) Bear in mind that if the front of the dampness edge is not generally horizontal the cause of the wetting could also be from elsewhere - although Fig 11 shows its all horizontal on that elevation.

p19 - I did think the interpretation of one set of samples in the rainy and the dry seasons was methodologically insufficient to conclude that free-thaw cycles were not significant re. brick damage - this should be recharacterised as needing further study/research, especially as here you are quantifying your results against rainfall which you have already said is not the source of the dampness - you need to redo this with reference to groundwater conditions and rising damp. Furthermore, any experimental results are never isomorphic with action/damage in the field, so I am not convinced by your comparison made in 7.2 b) especially as flaking etc can be caused by freeze-thaw action.

on p22(pdf) you make the correct claim about not using waterproofing agents but then go on to say that "it is highly unsuitable to cover masonry surfaces with lime" and reference one article on stress histories in sandstone as created by various things including lime renders - however many building conservation treatments include the use permeable lime washes and

renders as sacrificial coatings especially as they are seen to help salts crystallize out in them and not the stone substrate, which is something you do not address. Of course, whether you would want to cover historic brickwork in lime is another matter.

p22 (pdf) The "anti-weathering strategy" is the weakest part of the article.

You opt for the convention of installing French drains to mitigate the rising damp but then suggest removing rows of bricks and extant lime/clay mortar beds/pointing as shown in Fig 12 - historic in themselves - perhaps using the 'total cut' method, and replacing them with "waterproof materials" such as "steel [48], lead plates, bitumen-based membranes, polyethylene [49], and polyester-based membranes." This is dramatic stuff, however your ref. re. steel is slightly incorrect as the authors cited concluded that: "The groundwater level should be checked; if the level is likely to be higher than where the blocking of capillary suction is planned, the blocking should be watertight, e.g. not steel plates". This proviso applies here to all the materials suggested and needs to be built into any engineering works as proposed (but whihc you do not assess).

Of course what you propose is damaging to the cultural heritage you are trying to protect - as Doehne et al. note - ie. wire sawing out double rows of bricks and mortar at 2 different heights is very invasive as well as expensive and not without risk.

However, even though you suggest replacing the jointing with impermeable materials and old bricks with new more dense impermeable bricks fired at 1100 degrees C (p23 pdf) you also suggest the use of sacrificial bricks which I think sits oddly with the idea of the impermeable double layer, no? Why is this necessary?

So I'm not convinced you have thought this through quite enough from an engineering or conservation perspective and think these suggestions need revisiting and given more thought and properly phased in experimental regimes.

Furthermore you should be at least quantifying whether the extant pointing is impermeable or not - and if so first setting up a test area and replacing it with softer lime mortar to observe if it mitigate the effects of any decay.

You should also be proposing to set up other test areas to trial French drains, both with and without colour-matched sacrificial lime coats and with and without replacing the extant pointing with softer lime mortars ie. all standard practices. Only after these have failed to mitigate the effects of the rising damp should tests then look to more dramatic interventions, but also made first in controlled test sections.

Obviously desalination as proposed in the article to take place once your impermeable barrier has been installed but it had to eb thorough with poultice materials well matched to the substrate and salts being treated so this also needs more assessment than apparent in the article.

All of this is well within the scope of a section on further research and I suggest this as a way out of otherwise appearing to have rushed into unsubstantiated claims (p6 pdf) about having successfully 'safeguarding' the blue bricks.

Reviewer: 2

Comments to the Author(s)

Overall this is an interesting study, addressing a relatively little-researched brick type. The authors have gathered a lot of data across various methods. However, there are issues with this manuscript, in particular in the methodology and associated conclusions, which make it unsuitable for publication in its current state.

The authors mention three different sample types, but throughout the manuscript it is not made clear which samples are analysed. While the reader can work it out with some effort, it would be easier if the authors provided a clear overview of which samples was used for which test and

how these findings are tied together. As it stands the study presented here is rather fragmented. This therefore undermines the credibility of the conservation suggestions made by the authors. The methodology is not always clearly explained and justified, which again makes it more difficult to follow the narrative laid out by the authors. In fact, some methodologies, such as the repeated heating of blocks, are not justified at all which undermine the interpretations set out in the discussions. Towards the end of the manuscript there are quite a lot of frustrating statements, especially when the authors introduce environmental conditions without providing any sound environmental assessments.

P4

The authors need to embed further literature in their introduction. The second paragraph in particular, which describes the firing process for blue bricks, does not contain any further references, and lacks in detail such as firing temperature etc. There does seem to be considerable overlap between traditional red clay brick production (as seen widely around Europe) and blue brick production, yet the authors do not clearly differentiate this. Why not use more of the available literature on red bricks?

P5

L7 - 18 this paragraph is very anecdotal - give more examples and theoretical grounding.
L21- P6L14 There are two issues with this paragraph: [1] the examples cited are predominantly on European red brick, yet the authors have already stated that the blue brick is different - this has not been worked into the narrative presented here and [2] the studies are cited but there is no summary of their findings. So it is not clear how useful these studies are in providing the theoretical framework for the study presented here.

P6

Where are the samples in Fig 2 located in Fig 1?

L18 - 47 Again, there is little citation of the wider literature. It is also unclear what the authors mean with 'this study not only safeguards brick masonry...'

P7

L15: This needs a reference

L31: Define 'great preservational value' - are the authors talking about the value placed on the heritage by the local community?

L59: What evidence do the authors have for the height of the wetting zone?

P8

L5-7: This is a rather vague statement; do these classifications refer directly to the blue brick or is this simply the general list provided by ICOMOS?

L25: 'The samples belong to two sub-types according to the sampling parts:' - it is unclear what the authors mean by this

L21-33: The sampling of the bricks needs to be explained in much more detail. It is unclear where the samples were taken, how they were selected and how representative they are. The authors have already flagged up the importance of rising damp, yet do not provide detail on the location of the samples within the wider wall.

L35 - 38: How were these samples selected and obtained?

L41 - 55: The origins and potential weathering history of these bricks seem dubious. Do the authors have further information about their production history and origins?

P9:

L13: Why are they indispensable? Make sure that statements like these are properly explained.

L16: Which samples are the authors referring to? Sample set 1 or 3?

P10:

L12 - 18: Why do the authors not take the elasticity of the material into account when considering the potential damage of frost action?

L 24 - 32: Sample size and drying methods are not justified.

L44: What is 'pure water'? Distilled or deionised water?

L39 - 57: Why did the authors change the sample size from the one used for the mercury intrusion test? What were the time intervals and how were they determined?

P11:

L8: The use of the word 'inevitable' is very odd – the authors really need to be far more thorough in justifying all methods.

L13: Why 0.5 MPa/s? What does this load represent?

L18: This is a very small sample size compared to the original size of the bricks. How do the authors translate these findings to the behaviour of the bricks in situ?

L25 – 60: This whole section is poorly justified. There seems to be no correlation between the cyclical / repeated firing and the production of the original bricks. The authors acknowledge that firing leads to changes in the brick's mineral phases, yet this is not taken into account during repeated incremental firing. This repeated firing could have altered the bricks sufficiently to influence any further testing.

P12

L25: Why these salts? Are these the ones found in situ?

L28-41: Again, the methods are poorly justified here – why the 2 hour cooling phase? Why not 1 hour or 12 hours? What salts were used in the solution? (this is not clear from the preceding paragraph). There is no indication of use of control samples.

L46 – 58: What is 'pure water'? Which samples were used? What would the implications of their potential weathering history be?

P13:

L 31-33: 'Gypsum is the product...atmospheres' It is unclear what the function of this sentence is here.

L39-47: The heterogeneity of the bricks stated here undermines the validity of the general conclusions drawn until now, and the general conclusions of the study. The authors need to address this issue clearly.

P14

L7: The heterogeneity of the bricks is not translated in the results presented in table 3.

L14: What are 'ordinary rocks'? L14 – 24 Make little sense to the reader.

L27 – 38: The authors have switched back to general observations, yet do not address the variability of the bricks!

L56-59 / Table 4: The results presented in the text and in the table are not correlated well – it is unclear how the (large!) % ranges presented in the text relate to the results presented in the table.

P15

L4: Which sample?

L20-24: The authors report that the bricks contain anywhere between 22-59% small pores in total connected porosity, which would fall well within the bimodal pore size distribution category. They therefore exclude their own study samples here!

L54: The classifications make little sense when presented here in the text.

P16

L10: Again, this statement ignores the heterogeneity of the bricks

L14: What constitutes an 'ordinary' or 'even porous' rock?

L19: Rising damp is strongly connected with temperatures, and thus evaporation rates. It is not just a function of the capillarity of the bricks. The authors need to take into account the environmental conditions of the sample sites, and also provide actual measurements to justify their claims of capillary rise heights.

L36 – 52: The authors give the impression here that their samples are drawn from a varied pool of bricks, with large variability in the original material as well as production method. It is therefore difficult to justify their interpretations considering the limited sampling they have undertaken and the variability in the nature of the samples.

P17

L4-27: See my previous point regarding cyclical and incremental testing vs the actual firing method used in the production of these bricks.

L36: What is classified as 'damage'? Small cracks or structural failure? Pitting of the surface? The graphs do not clearly indicate when the sample is damaged – is it when the line stops?

L48 – 58: Can the authors relate the variability in porosity to the response of the block to the salt weathering?

P18:

L14: In all samples? Are the results pooled? This again ignores the variability in the samples.

L24 – 31: Which samples? The authors are very vague in their statements here

L43-48: This is not clear from the presented results – the freeze thaw tests indicate macro-cracking. The authors do not provide detailed observations of salt weathering across the study site, nor do they provide sufficient information on diurnal temperature fluctuations throughout the year to discard freeze thaw weathering.

L58 – 60: This is the first time the authors introduce clay cores in the walls, which changes the potential behaviour of the bricks.

P19 and P20

This is the first time the authors take environmental conditions into account, and provide little in the way of measurements and observations of justify their statements. The study should be rewritten and restructured with environmental conditions taken into account for the authors to justify making such statements.

P20

L18 – 43: Freeze thaw damage effectiveness is dependent on repeated above and below zero temperatures, more so than moisture content. The authors have not measured temperatures. Therefore, their statements regarding the importance of freeze thaw are not justified.

L46: Frost heave is not the same as frost-induced cracking. Please consult the literature on ‘frost heave’

P21

L3-20: The authors invalidate their own findings, and admit the issues that I have already flagged up in this review. So, how valid are the conclusions drawn from this study? In my view they are not justified.

In light of the problems flagged up regarding the variability of the material in the test results and the discrepancy between in situ conditions, environmental conditions and the laboratory samples it is very questionable whether or not the ‘anti-weathering strategies’ are justified. In my view they are not.

Author's Response to Decision Letter for (RSOS-191008.R0)

See Appendix A.

RSOS-200058.R0

Review form: Reviewer 1

Is the manuscript scientifically sound in its present form?

Yes

Are the interpretations and conclusions justified by the results?

Yes

Is the language acceptable?

Yes

Do you have any ethical concerns with this paper?

No

Have you any concerns about statistical analyses in this paper?

No

Recommendation?

Accept with minor revision (please list in comments)

Comments to the Author(s)

Thank you for responding to many of the comments I previously made.

Although you have moderated your claims I do think you still need to strongly recast the work as part of a larger study ie. as a pilot project or phase 1 of a study to describe the deterioration mechanisms affecting the blue bricks, rather than assert here without sufficient data that salts are the only major factor.

This is because there are still 2 important problems:

1) Both I and the other reviewer have misgivings about sample size, non-homogeneity of the material composition vs. homogeneity of samples, and the environmental testing/ data used (or not used) etc, which impact the assertion that salt crystallization is the major deterioration factor for the bricks (cf. 7.1)

It may well be the case that salts are the biggest factor but you have not given enough weight to the effects of freeze-thaw action which, by the now added figures from the local Meteorology Bureau, suggest that freezing temperature conditions occur for 6 months (<0 centigrade October-April) and by extrapolating from the Bureau's rainfall figures, significant to moderately significant rain occurs for at least 3 months that overlap those freezing temperatures (October, November and March).

The fact that water uptake by the bricks is reported and that rising damp is observed up to 3m, and that the PSD profile benefits both salt crystallization and freeze/thaw, all suggest freeze/thaw needs further investigating as you acknowledge. This means the assertions in 7.1 etc need modifying.

2) The 2nd point is that the 'coving' described is attributed to gypsum formation/solubility and/or mineral inhomogeneity in the bricks themselves.

However although the 'brick-mortar' interface is mentioned as contributing there is no description of what the mortar actually is. I did ask for this to be reported in my first review.

Fig 2 c & d shows what appears to be a white mastic as mortar/pointing (old mastic are often an oil mixed with CaCO₃ eg. chalk) and this seems to be proud of the surface suggesting that the coving is determined by water egress from the brick faces and not joints. This means that any soluble salts will of course egress out from the brick faces too, rather than the joints. So I am not convinced the authors have fully described the cause of the coving here and I suggest they look at the permeability of the mortar compared with the brick. There is much literature on this kind of scaling and its relation to bedding/pointing (especially with regard to stone decay).

Further points:

In terms of mitigating the effects of both salt and water egress, as I said before testing needs to be made with both softer lime pointing (but being mindful of local traditions) and also lime washes (again mindful of traditions). Lime washes as sacrificial coating in principle push the zone of salt efflorescence further forward from the brick faces and these could also be tested.

Desalination methods also need proper investigating rather than just the assertion of specific products by citation. Poulticing can be appropriate but only if the wall is isolated from the source of soluble salts - otherwise its pointless.

Decision letter (RSOS-200058.R0)

Dear Dr Zhang,

The Subject Editor assigned to your paper ("Characteristics and weathering mechanisms of the traditional Chinese blue brick from the Ancient City of Ping Yao") has now received comments from reviewers. We would like you to revise your paper in accordance with the referee and Associate Editor suggestions which can be found below (not including confidential reports to the Editor). Please note this decision does not guarantee eventual acceptance.

Please submit a copy of your revised paper before 23-May-2020. Please note that the revision deadline will expire at 00.00am on this date. If we do not hear from you within this time then it will be assumed that the paper has been withdrawn. In exceptional circumstances, extensions may be possible if agreed with the Editorial Office in advance. We do not allow multiple rounds of revision so we urge you to make every effort to fully address all of the comments at this stage. If deemed necessary by the Editors, your manuscript will be sent back to one or more of the original reviewers for assessment. If the original reviewers are not available we may invite new reviewers.

When submitting your revised manuscript, you must respond to the comments made by the referees and upload a file "Response to Referees" in "Section 6 - File Upload". Please use this to document how you have responded to each of the comments, and the adjustments you have made. In order to expedite the processing of the revised manuscript, please be as specific as possible in your response.

- Ethics statement

- Data accessibility

If you wish to submit your supporting data or code to Dryad (<http://datadryad.org/>), or modify your current submission to dryad, please use the following link:
<http://datadryad.org/submit?journalID=RSOS&manu=RSOS-200058>

- Competing interests

- Authors' contributions

- Acknowledgements

- Funding statement

on behalf of R. Kerry Rowe (Subject Editor)
openscience@royalsociety.org

Associate Editor Comments to Author:
Comments to the Author:

Thank you for submitting your resubmission. Following peer review, we've received one report on your manuscript.

The referee stated that there are still two important problems that need addressing. Please ensure

that you address all these concerns appropriately upon resubmission, and provide a detailed point-by-point response.

Reviewer comments to Author:

Reviewer: 1

Comments to the Author(s)

Thank you for responding to many of the comments I previously made.

Although you have moderated your claims I do think you still need to strongly recast the work as part of a larger study ie. as a pilot project or phase 1 of a study to describe the deterioration mechanisms affecting the blue bricks, rather than assert here without sufficient data that salts are the only major factor.

This is because there are still 2 important problems:

1) Both I and the other reviewer have misgivings about sample size, non-homogeneity of the material composition vs. homogeneity of samples, and the environmental testing/ data used (or not used) etc, which impact the assertion that salt crystallization is the major deterioration factor for the bricks (cf. 7.1)

It may well be the case that salts are the biggest factor but you have not given enough weight to the effects of freeze-thaw action which, by the now added figures from the local Meteorology Bureau, suggest that freezing temperature conditions occur for 6 months (<0 centigrade October-April) and by extrapolating from the Bureau's rainfall figures, significant to moderately significant rain occurs for at least 3 months that overlap those freezing temperatures (October, November and March).

The fact that water uptake by the bricks is reported and that rising damp is observed up to 3m, and that the PSD profile benefits both salt crystallization and freeze/thaw, all suggest freeze/thaw needs further investigating as you acknowledge. This means the assertions in 7.1 etc need modifying.

2) The 2nd point is that the 'coving' described is attributed to gypsum formation/solubility and/or mineral inhomogeneity in the bricks themselves.

However although the 'brick-mortar' interface is mentioned as contributing there is no description of what the mortar actually is. I did ask for this to be reported in my first review.

Fig 2 c & d shows what appears to be a white mastic as mortar/pointing (old mastic are often an oil mixed with CaCO₃ eg. chalk) and this seems to be proud of the surface suggesting that the coving is determined by water egress from the brick faces and not joints. This means that any soluble salts will of course egress out from the brick faces too, rather than the joints. So I am not convinced the authors have fully described the cause of the coving here and I suggest they look at the permeability of the mortar compared with the brick. There is much literature on this kind of scaling and its relation to bedding/pointing (especially with regard to stone decay).

Further points:

In terms of mitigating the effects of both salt and water egress, as I said before testing needs to be made with both softer lime pointing (but being mindful of local traditions) and also lime washes (again mindful of traditions). Lime washes as sacrificial coating in principle push the zone of salt efflorescence further forward from the brick faces and these could also be tested.

Desalination methods also need proper investigating rather than just the assertion of specific

products by citation. Poulting can be appropriate but only if the wall is isolated from the source of soluble salts - otherwise its pointless.

Author's Response to Decision Letter for (RSOS-200058.R0)

See Appendix B.

RSOS-200058.R1 (Revision)

Review form: Reviewer 1

Is the manuscript scientifically sound in its present form?

Yes

Are the interpretations and conclusions justified by the results?

Yes

Is the language acceptable?

Yes

Do you have any ethical concerns with this paper?

No

Have you any concerns about statistical analyses in this paper?

No

Recommendation?

Accept as is

Comments to the Author(s)

I am glad to see that the earlier claims for the results from the work have been significantly modified and the nature of the study as a pilot/phase 1 of a potentially larger project is much clearer, and with the further work necessary to develop more definitive conclusions signposted. I was also glad to see a note on the mortar permeability and its potential as being active in some of the decay of the bricks.

I think you could briefly describe the mortar and change the caption of Fig 2 and identify the mortar (eg. in 2c) and relate this back to 6.2.2 and maybe include comparative curves for the mortar and brick and then describe how the scaling could be related to the less porous/more impermeable mortar.

Otherwise I am happy to pass this 3rd version with only some minor corrections to the English needed here and there. Thank you.

Decision letter (RSOS-200058.R1)

Dear Dr zhang:

On behalf of the Editors, I am pleased to inform you that your Manuscript RSOS-200058.R1 entitled "Characteristics and weathering mechanisms of the traditional Chinese blue brick from the Ancient City of Ping Yao" has been accepted for publication in Royal Society Open Science subject to minor revision in accordance with the referee suggestions. Please find the referees' comments at the end of this email.

The reviewers and Subject Editor have recommended publication, but also suggest some minor revisions to your manuscript. Therefore, I invite you to respond to the comments and revise your manuscript.

- Ethics statement

- Data accessibility

<http://datadryad.org/submit?journalID=RSOS&manu=RSOS-200058.R1>

- Competing interests

- Authors' contributions

- Acknowledgements

- Funding statement

Because the schedule for publication is very tight, it is a condition of publication that you submit the revised version of your manuscript before 03-Jul-2020. Please note that the revision deadline will expire at 00.00am on this date. If you do not think you will be able to meet this date please let me know immediately.

Supplementary files will be published alongside the paper on the journal website and posted on the online figshare repository (<https://figshare.com>). The heading and legend provided for each supplementary file during the submission process will be used to create the figshare page, so

please ensure these are accurate and informative so that your files can be found in searches. Files on figshare will be made available approximately one week before the accompanying article so that the supplementary material can be attributed a unique DOI.

on behalf of Prof R. Kerry Rowe (Subject Editor)
openscience@royalsociety.org

Associate Editor Comments to Author

Thank you for so constructively engaging with the reviewers' feedback at all stages of the revision process. Only a few minor changes are recommended before acceptance (see the referee's comments); however, we would like you to consider seeking advice from a native speaker of English or a language editing service (<https://royalsociety.org/journals/authors/benefits/language-editing/>) to tidy up a few remaining matters. Otherwise, thank you for your support of the journal.

Reviewer comments to Author:

Reviewer: 1

Comments to the Author(s)

I am glad to see that the earlier claims for the results from the work have been significantly modified and the nature of the study as a pilot/phase 1 of a potentially larger project is much clearer, and with the further work necessary to develop more definitive conclusions signposted. I was also glad to see a note on the mortar permeability and its potential as being active in some of the decay of the bricks.

I think you could briefly describe the mortar and change the caption of Fig 2 and identify the mortar (eg. in 2c) and relate this back to 6.2.2 and maybe include comparative curves for the mortar and brick and then describe how the scaling could be related to the less porous/more impermeable mortar.

Otherwise I am happy to pass this 3rd version with only some minor corrections to the English needed here and there. Thank you.

Author's Response to Decision Letter for (RSOS-200058.R1)

See Appendix C.

Decision letter (RSOS-200058.R2)

Dear Dr zhang,

It is a pleasure to accept your manuscript entitled "Characteristics and weathering mechanisms of the traditional Chinese blue brick from the Ancient City of Ping Yao" in its current form for publication in Royal Society Open Science.

Please ensure that you send to the editorial office individual files for each figure and table included in your manuscript. You can send these in a zip folder if more convenient. Failure to provide these files may delay the processing of your proof. You may disregard this request if you have already provided these files to the editorial office.

on behalf of Prof R. Kerry Rowe (Subject Editor)
openscience@royalsociety.org

Appendix A

Responses to Reviewers' Comments

Zhongjian Zhang

China University of Geosciences, Beijing, China.

zhangzhongjian@cugb.edu.cn

Acknowledgement The authors are grateful to the anonymous reviewer for a careful checking of the details and for helpful comments that improved this paper.

Responses to Reviewer #1:

Reviewer: 1

Comments to the Author(s)

Generally a good article where the authors have understood that measuring the concentration of salt in the substrate is only one part of the issue, since substrate characteristics determine the its resistance to salt decay. However there is less detail on the severity and frequency of environmental fluctuations which are equally important in determining rates of salt damage and this reviewer feels this should be addressed more thoroughly.

Comments on the text:

the term 'folk house' needs definition.

p4(pdf) Ln44+ - useful to know at what temperatures the adobe blocks were fired. In Europe there are both low temp. and high temp. bricks and the process of their manufacture is not as different as claimed with both sharing processes of controlled mixing, moulding, natural drying and firing.

p5(pdf) presents a list of publications on red brick conservation - which is not the subject under discussion and could be made much shorter to help the point made in the next paragraph on p6 (pdf) that there are few studies on blue bricks. Where the mineral compositions are found to be similar this should all be detailed in the relevant section on p13 (pdf).

p6 (pdf) Ln 39+ there is a bold and ultimately incorrect claim about 'safeguarding' blue bricks that needs qualification (see later).

the 'average' annual figures given on p7(pdf) are not useful and its the min/max ranges and length of exposure that are important. For example year long microclimate monitoring would show if there were particularly long and deleterious wetting-drying and freeze-thaw cycles etc esp. as on p17 (pdf) you say that

"The repeated change in environmental factors around the walls (e.g., humidity and temperature) leads to increasing amounts of salt left on the wall surfaces as well as in the brick pores, thus the tension in the brick pores continues to increase"

but do not state where this information on rapid environmental change comes from.

Given the 'averages' given before, the description of weathering types is extremely brief and conflated with rising damp. This section needs a little more explanation relating to particular mechanisms observed in the locale. It incorrectly lists the effects of weathering - contour scaling, flaking etc - as 'weathering types'.

The analyses is where you exhibit greatest confidence and the results confirm standard interpretation eg. greater water transport when smaller capillary radii and that distribution is never regular. Within the given ranges of 0.1 to 5 μm pore radii its interesting to note if you can confirm Winkler's(1994) findings that capillaries $\leq 0.1 \mu\text{m}$ are generally unable to absorb water (so the greater % of their distribution, the lower the SSI)?

p14 (pdf) what you haven't reported on is the type of mortar used between the bricks - this is essential to understand as if the mortar is more impermeable than the bricks then the concave decay described here might be caused by the harder mortar (it looks like a mastic in Fig 2) rather than just the differences in manufacturing mixes to which you attribute it. I think this omission needs looking at before publication - cf. definition of 'coving' as an erosion feature consisting in a single alveole developing from the edge of the stone block often caused by harder/more impermeable pointing on p28 of https://www.icomos.org/publications/monuments_and_sites/15/pdf/Monuments_and_Sites_15_ISCS_Glossary_Stone.pdf

re. p16(pdf) 6.2.3 - the speed and height variation in capillary rise are to be expected as the bricks are non-homogeneous in composition - Camuffo (2014) has some interesting work in this area.

re. p17(pdf) re. TRM/archaeomagnetic dating - were analyses made to identify any local spikes or rapid changes in geomagnetic activity etc in the 14-17th C period? This needs consideration given the claim that 'the original maximum firing temperatures of the brick samples are identified' in your covering letter.

re. 7.1 - this needs to take account that rising damp causes the mobilisation of soluble salts in the bricks leading to their crystallization. Crystallization should be differentiated more clearly between efflorescence and subfluorescence - its the latter, as you identify, that causes pore disruption and substrate damage.

p18 (pdf) Bear in mind that if the front of the dampness edge is not generally horizontal the cause of the wetting could also be from elsewhere - although Fig 11 shows its all horizontal on that elevation.

p19 - I did think the interpretation of one set of samples in the rainy and the dry seasons was methodologically insufficient to conclude that free-thaw cycles were not significant re. brick damage - this should be recharacterised as needing further study/research, especially as here you are quantifying your results against rainfall which you have already said is not the source of the dampness - you need to redo this with reference to groundwater conditions and rising damp. Furthermore, any experimental results are never isomorphic with action/damage in the field, so I am not convinced by your comparison made in 7.2 b) especially as flaking etc can be caused by freeze-thaw action.

on p22(pdf) you make the correct claim about not using waterproofing agents but then go on to say that "it is highly unsuitable to cover masonry surfaces with lime" and reference

one article on stress histories in sandstone as created by various things including lime renders - however many building conservation treatments include the use permeable lime washes and renders as sacrificial coatings especially as they are seen to help salts crystallize out in them and not the stone substrate, which is something you do not address. Of course, whether you would want to cover historic brickwork in lime is another matter.

p22 (pdf) The "anti-weathering strategy" is the weakest part of the article.

You opt for the convention of installing French drains to mitigate the rising damp but then suggest removing rows of bricks and extant lime/clay mortar beds/pointing as shown in Fig 12 - historic in themselves - perhaps using the 'total cut' method, and replacing them with "waterproof materials" such as "steel [48], lead plates, bitumen-based membranes, polyethylene [49], and polyester-based membranes." This is dramatic stuff, however your ref. re. steel is slightly incorrect as the authors cited concluded that: "The groundwater level should be checked; if the level is likely to be higher than where the blocking of capillary suction is planned, the blocking should be watertight, e.g. not steel plates". This proviso applies here to all the materials suggested and needs to be built into any engineering works as proposed (but whihc you do not assess).

Of course what you propose is damaging to the cultural heritage you are trying to protect - as Doehne et al. note - ie. wire sawing out double rows of bricks and mortar at 2 different heights is very invasive as well as expensive and not without risk.

However, even though you suggest replacing the jointing with impermeable materials and old bricks with new more dense impermeable bricks fired at 1100 degrees C (p23 pdf)

you also suggest the use of sacrificial bricks which I think sits oddly with the idea of the impermeable double layer, no? Why is this necessary?

So I'm not convinced you have thought this through quite enough from an engineering or conservation perspective and think these suggestions need revisiting and given more thought and properly phased in experimental regimes.

Furthermore you should be at least quantifying whether the extant pointing is impermeable or not - and if so first setting up a test area and replacing it with softer lime mortar to observe if it mitigate the effects of any decay.

You should also be proposing to set up other test areas to trial French drains, both with and without colour-matched sacrificial lime coats and with and without replacing the extant pointing with softer lime mortars ie. all standard practices. Only after these have failed to mitigate the effects of the rising damp should tests then look to more dramatic interventions, but also made first in controlled test sections.

Obviously desalination as proposed in the article to take place once your impermeable barrier has been installed but it had to eb thorough with poultice materials well matched to the substrate and salts being treated so this also needs more assessment than apparent in the article.

All of this is well within the scope of a section on further research and I suggest this as a way out of otherwise appearing to have rushed into unsubstantiated claims (p6 pdf) about having successfully 'safeguarding' the blue bricks.

Comment 1:

Comments to the Author(s)

Generally a good article where the authors have understood that measuring the concentration of salt in the substrate is only one part of the issue, since substrate characteristics determine the its resistance to salt decay. However there is less detail on the severity and frequency of environmental fluctuations which are equally important in determining rates of salt damage and this reviewer feels this should be addressed more thoroughly..

Response 1:

Thank you very much for reviewing our manuscript! The authors appreciate your comments and insightful suggestions to our manuscript. According to your suggest, the authors made a thorough revision of this manuscript. The more detailed environmental information of the study area is also added and the the relevant part of are addressed to have a stronger link between the results and the environment fluctuations.

Comment 2:

Comments on the text:

the term 'folk house' needs definition.

Response 2:

Thank you very much for your suggestions. The “folk houses” means the house privately owned by citizens, the “folk houses” are replaced by “private houses” throughout the manuscript to avoid misunderstanding.

Comment 3:

p4(pdf) Ln44+ - useful to know at what temperatures the adobe blocks were fired. In Europe there are both low temp. and high temp. bricks and the process of their manufacture is not as different as claimed with both sharing processes of controlled mixing, moulding, natural drying and firing.

Response 3:

Thanks for your comment. Indeed the firing temperature can lead to the difference in properties of bricks. There are low and high firing temperatures of red bricks, which can cause the difference in mineral phases. Therefore, this part has been rewritten and the firing temperatures of red and blue bricks are specified.

Page 3-4

“The firing temperatures, which directly effect the textural evolution, vitrification, pore size distribution (PSD) of the bricks, and hence the physico-mechanical properties and the weathering resistance, also show some differences between red and blue bricks according to the literature [1,3-5,7]). For example, Elert et al. [5] studied the influence of firing temperature on bricks made by two typical clays, and found that the low-temperature-fired (700-800 °C) generally present inferior PSD, strength, and weathering resistance including freeze-thaw and salt crystallization, compared with the high-temperature-fired (over 1000 °C) bricks. Lopez-Arce et al. [4] found that the firing temperature is directly related to bricks’ properties by controlling the phase changes of minerals, e.g., the new mineral phases (gehlenite and diopside) were formed at the expense of calcite and dolomite above 800 °C. Cultrone et al. [7] found the bricks used for construction of ‘Triangul Bastion’, Riga (Latvia) were fired at around 900 °C, they also found the difference in color corresponds to difference in minerals: yellow color bricks were characterized by quartz, diopside and minor amounts of feldspar; the red bricks were much richer in quartz and feldspar. The data regarding firing

temperatures of Chinese blue bricks are limited in literature, Shu et al. [1] studied fourteen blue and red brick samples derived from four modern buildings (constructed between 1866 and 1935) in Shanghai, China and reported that the maximum firing temperatures of blue bricks are ranging from 600 to 800 °C. The distinctions of brick-making environments (i.e. oxygen, temperature) should produce the blue bricks unique characteristics that are worth of investigation.”

Comment 4:

p5(pdf) presents a list of publications on red brick conservation - which is not the subject under discussion and could be made much shorter to help the point made in the next paragraph on p6 (pdf) that there are few studies on blue bricks. Where the mineral compositions are found to be similar this should all be detailed in the relevant section on p13 (pdf).

Response 4:

Thank you for your suggestion. According to our suggestion, the authors has shorten the content on red brick conservation.

Page 2-3

“Most existing studies have focused on red bricks and have achieved considerable results regarding the influence of the weathering factors and mechanisms such as water impregnation during rainy seasons and water evaporation during dry seasons [2], the raw clay mineral compositions and the mineralogical and textural evolution during the firing process [3-5], the physical and mechanical properties [6], as well as the weathering behaviour [7]. Compared with the widely studied red bricks, the scientific understanding to the blue bricks haven't

receive enough attention, even though the brick-making methods of the traditional Chinese blue brick are very different from the methods used in making European red bricks and the properties may thus be different [1].”

Comment 5:

p6 (pdf) Ln 39+ there is a bold and ultimately incorrect claim about 'safeguarding' blue bricks that needs qualification (see later).

the 'average' annual figures given on p7(pdf) are not useful and its the min/max ranges and length of exposure that are important. For example year long microclimate monitoring would show if there were particularly long and deleterious wetting-drying and freeze-thaw cycles etc esp. as on p17 (pdf) you say that

"The repeated change in environmental factors around the walls (e.g., humidity and temperature) leads to increasing amounts of salt left on the wall surfaces as well as in the brick pores, thus the tension in the brick pores continues to increase" but do not state where this information on rapid environmental change comes from.

Given the 'averages' given before, the description of weathering types is extremely brief and conflated with rising damp. This section needs a little more explanation relating to particular mechanisms observed in the locale. It incorrectly lists the effects of weathering - contour scaling, flaking etc - as 'weathering types'.

Response 5:

Thanks. Yes, this claim is too bold and ultimately incorrect, since the main focus of this paper is to investigate the main factors leading to the weathering phenomenon and through field investigation and the element tests. The main contribution of this paper is to provide understanding of the properties of the blue brick and to identify the main weathering factors and mechanisms. Then the authors made some discussion on the possible approaches to

protect the blue brick relics in Ping Yao based on the results, yet the validation on protection methods are not included in current paper. Therefore, the authors changed this claim.

Indeed, the more important part of the environment is the range and the max/min value as well as the variation periods in a year. The authors changed this part by adding the max/min value and the corresponding seasons to help with the discussion and understanding of the factors that lead to the weathering of the blue brick studied. The authors also corrected the incorrect usage of the term “weathering types”.

Page 4-5

“Since there are many historic Chinese architectural structures constructed of blue brick, this study not only provides understandings and possible approaches on the conversation of brick masonry in the Ancient City of Ping Yao, which is of great importance, but also relates the possible implications of those results to other similar cases.”

Page 5-6

“The environmental information provided by the Ping Yao Meteorological Bureau indicates that the average annual insolation reaches 2433.2 hours. The average annual temperature is 10.4 °C, ranging from -5.4 °C in January to 24.2 °C in July. The temperature generally below 0 °C from the Mid-October to the Mid-April next year. The average rainfall is 415.5 mm, falling mainly in the summer months. The rainfall from July to September accounts for 58.8 % of total annual rainfall amount, while the rainfall from December to February only accounts for 3.5 % of the total annual rainfall. The annual relative humidity is 58 %. The average wind speed is 2.2 m/s, being higher in spring and winter and lower in summer and autumn. The prevailing wind is mainly from the southwest but from the northeast between June and September.”

Comment 6:

The analyses is where you exhibit greatest confidence and the results confirm standard interpretation e.g. greater water transport when smaller capillary radii and that distribution is never regular. Within the given ranges of 0.1 to 5 μm pore radii its interesting to note if you can confirm Winkler's(1994) findings that capillaries $\leq 0.1 \mu\text{m}$ are generally unable to absorb water (so the greater % of their distribution, the lower the SSI)?

Response 6:

Thank you for your comment. The authors agree with your opinion. As for the relationship between Winkler's finding and the SSI, the authors hold the point of view that they are not contradictory. The Winkler's finding is about the water absorption ability, while the SSI is an index describing the easiness of being broken by salt crystallization. Yes, the micropores (generally $\leq 0.2 \mu\text{m}$) is generally not easy to absorb water, especially for the pores $\leq 0.1 \mu\text{m}$, but when given enough time the pores is still able to get some water, and a small amount of salt can exert great pressure on the walls due to the pore size and then break the pores easily.

Comment 7:

p14 (pdf) what you haven't reported on is the type of mortar used between the bricks - this is essential to understand as if the mortar is more impermeable than the bricks then the concave decay described here might be caused by the harder mortar (it looks like a mastic in Fig 2) rather than just the differences in manufacturing mixes to which you attribute it. I think this omission needs looking at before publication - cf. definition of 'coving' as an erosion feature consisting in a single alveole developing from the edge of the stone block often caused by harder/more impermeable pointing on p28 of

https://www.icomos.org/publications/monuments_and_sites/15/pdf/Monuments_and_Sites_15_ISCS_Glossary_Stone.pdf.

Response 7:

Thanks for your comment. Indeed, it is possible that a differential weathering possibly due to inhomogeneities in physical or chemical properties can lead to the “coving” effects as described in the ICOMOS-ISCS. In this case the erosion should start from the brick-mortar interface (namely the edge of the bricks), as shown in Fig. 2a. The authors included this in our revised manuscript as follow:

Page 12

“Greater amounts of gypsum and calcite are found in weathered samples than in compact samples, especially in samples from concave parts. Gypsum is the product of the reaction of sulfur compounds in polluted atmospheres (e.g. $\text{Ca}^{2+} + \text{SO}_2 \rightarrow \text{CaSO}_4$) [24,25]. The clay mineral content in samples from greatest to least is concave parts > convex parts > compact samples, while the quartz content levels are in the reverse order. The difference in clay and quartz content levels across bricks is likely due to discrepancies in raw clay minerals and brick-making techniques (e.g. uniformity of clay mixture, local sintering temperatures). The inhomogeneities in mineral composition result in different weathering resistance both among different bricks and also in the different parts of the same brick [8,26]. In light of this finding and the explanation in the ICOMOS-ISCS [8], the erosion of the brick-mortar interface in Fig. 2 is also likely related to the alveolization (or coving) that caused by different physical and chemical properties between the mortar and the brick.”

Comment 8:

re. p16(pdf) 6.2.3 - the speed and height variation in capillary rise are to be expected as the bricks are non-homogeneous in composition - Camuffo (2014) has some interesting work in this area.

Response 8:

Thank you for your suggestion. The authors agree with this suggestion and have added this interesting explanation in the manuscript.

Page 15

“Interestingly, the water absorption coefficients of different brick samples show variation in both rising speed and rising height, which may explain the difference in the rising height observed in different walls. The different speeds and heights in capillary rise may be a result of non-homogeneous composition of bricks [37].”

Comment 9:

re. p17(pdf) re. TRM/archaeomagnetic dating - were analyses made to identify any local spikes or rapid changes in geomagnetic activity etc in the 14-17th C period? This needs consideration given the claim that 'the original maximum firing temperatures of the brick samples are identified' in your covering letter.

Response 9:

Thanks for this comment. The author used the word “original” here to address the initial firing temperature. After seeing this comment, the authors checked the reference paper (Shu et al, 2017 [1]) and found that they also didn’t use the word. Therefore, the authors change the “original firing temperature” into “firing temperature” following the term in reference [1] throughout the cover letter and the manuscript to avoid causing such confusion.

Comment 10:

re. 7.1 - this needs to take account that rising damp causes the mobilisation of soluble salts in the bricks leading to their crystallization. Crystallization should be differentiated more clearly between efflorescence and subfluorescence - its the latter, as you identify, that causes pore disruption and substrate damage.

Response 10:

Thank you for your reminding. The authors apologize for this ambiguity caused. To avoid misunderstanding, the authors rewrote this part and specifically pointed out that it is the subfluorescence that caused the disruption and substrate damage.

Page 17-18

“Moreover, in dry seasons, due to a moisture reduction and/or a temperature decrease, the soluble salt begins to absorb water vapor and form salt crystals, such as hexahydrate and epsomite. The volume increase during salt crystallization (e.g. subfluorescence) often applies tension in the brick pores and thus causes pore disruption and substrate damage.”

Comment 11:

p18 (pdf) Bear in mind that if the front of the dampness edge is not generally horizontal the cause of the wetting could also be from elsewhere - although Fig 11 shows its all horizontal on that elevation.

Response 11:

Thanks, the authors agree with this. The authors also observed the front of the dampness edge is not generally horizontal. The purpose of Fig.11 is as an example that the practice of

smearing cement, clay, and plant fibres on the ancient brick in walls failed to prevent weathering of the ancient bricks.

Comment 12:

p19 - I did think the interpretation of one set of samples in the rainy and the dry seasons was methodologically insufficient to conclude that free-thaw cycles were not significant re. brick damage - this should be recharacterised as needing further study/research, especially as here you are quantifying your results against rainfall which you have already said is not the source of the dampness - you need to redo this with reference to groundwater conditions and rising damp. Furthermore, any experimental results are never isomorphic with action/damage in the field, so I am not convinced by your comparison made in 7.2 b) especially as flaking etc can be caused by freeze-thaw action.

Response 12:

Thank you very much for this comment. The authors re-cogitated the role of freeze-thaw according to the comment very carefully and drew the conclusion that it was improper to boldly claim the conclusion. The crystallization is a main weathering factor based on the experimental and field investigation. But the freeze-thaw cycles, at least from the current information, can't be claimed as a second weathering factor: first, the lab test of the blue brick samples show that the material is prone to freeze-thaw; second, the previous consideration that the freeze thaw cycles is not the main weathering factor because of the "limited rainfall and moisture content in winter" is not appropriate, since the freeze-thaw weathering also dependent on the PSD and the tensile strength of the material; in small pores, a small amount of water can also exert huge pressure after freezing and break the material in near region. Therefore, it is better to admit the influence of freeze-thaw cycles on the weathering of the blue brick, but to what extent and how dominant is it is still need further investigation on the

ground water condition and more field monitoring data. The author revised the relevant section and the new text is shown as follow:

Page 18-20

“7.2 Effect of the freeze-thaw cycles

The brick samples are all damaged after 10 freeze-thaw cycles, which denotes that the freeze-thaw cycles can contribute to the weathering of the Ping Yao ancient bricks. Frost damage could possibly be a combination of three principal theories [41]: the volumetric expansion (approximately 9 %) of water on freezing, the crystallization or "ice lens theory", and the hydraulic pressure theory. The freeze-thaw cycles are definitely detrimental to the preservation of the ancient brick from the laboratorial results. However, whether the freeze-thaw cycles are the dominant factor or not still need further study, since several inconsistencies are observed between the field observation and the laboratory results, as discussed below:

(a) Even though the lab experiment show significant damage caused by freeze-thaw cycles, the effect is not necessarily strong because of the low moisture content of the brick walls during cold season. To assess the influence of damp on the bricks, an IMKO HD2-Mobile Moisture Metre equipped with TRIME-PICO probes was used to measure the moisture content (MC) of the joint mortars between bricks. The MC tests were carried out at four different locations on June 20th (in the rainy season), 2013. In each location, 4 to 10 different spots ranging from 0.5-1.5 m above ground level were tested. The tests were repeated in the same locations on October 26th (in the dry season), 2013. On June 20th, the maximum MCs (%) of the four measurement locations were 28.49, 65.94, 27.82, and 35.94, respectively, with a mean value of 39.66. However, on October 26th, the corresponding data

were reduced to 6.10, 15.30, 7.52, and 10.89, respectively, with a mean value of 9.95 (a 75 % drop). The tested results are consistent with the weather conditions of the Ancient City of Ping Yao, i.e., the average rainfall in the winter only accounts for 3.5 % of the average annual rainfall of 415.5 mm in the county. Therefore, it seems that there is no enough moisture in bricks during cold to form ice in micro pores.

(b) The frost heaving damage modes in the experiment (Fig. 9) are not quite consistent with the modes observed on-site. In the laboratory, the typical frost heaving damage mode of the ancient bricks is macroscopic cracking, and the cleavages of different samples are at roughly equal levels (Fig. 9). The failure modes observed on-site include contour scaling and granular disintegration, which are more similar to the failure modes caused by salt crystallization; while the flaking observed on site is commonly a product of freeze-thaw weathering.

Therefore, it can be confirmed that the freeze-thaw is a factor accounting for the weathering of ancient blue brick in Ping Yao. Yet the questions of “how detrimental is the freeze-thaw” and “is the freeze-thaw a more or less dominant weathering factor compared with the salt crystallization” are still unclear and need to be further investigated in next step. To have a quantitative understanding, the moisture content in brick walls, rainfall condition, and underground water condition in wet and dry seasons need to be monitored at more locations; and then the lab-based freeze-thaw cycles should be conducted on the samples with certain moisture contents that based on the field results.”

Comment 13:

on p22(pdf) you make the correct claim about not using waterproofing agents but then go on to say that "it is highly unsuitable to cover masonry surfaces with lime" and reference one article on stress histories in sandstone as created by various things including lime renders - however many building conservation treatments include the use permeable lime washes and renders as sacrificial coatings especially as they are seen to help salts crystallize out in them and not the stone substrate, which is something you do not address. Of course, whether you would want to cover historic brickwork in lime is another matter.

Response 13:

Thanks for reminding. The authors apologize for citing a paper regarding the sandstone, although only a part of the content that the authors intent to use. The authors have consideration for aesthetic and protecting perspective, since this is a UNESCO site. The authors deleted the citation and the relevant expression. The new content is as follows.

Page 20

“The present protective covering is observed to be made by cement, clay and plant fires (Fig. 11) in Ping Yao. However, these covering materials failed to retard the weathering of the brick masonry. In reality, it is unsuitable to cover masonry surfaces with waterproofing materials (e.g., impermeable plasters and bitumen) [42]. The misuse of repair materials may itself damage the cultural relics, and is considered improper artificial repair.”

Comment 14:

p22 (pdf) The "anti-weathering strategy" is the weakest part of the article.

You opt for the convention of installing French drains to mitigate the rising damp but then suggest removing rows of bricks and extant lime/clay mortar beds/pointing as shown in Fig 12 - historic in themselves - perhaps using the 'total cut' method, and replacing them with

"waterproof materials" such as "steel [48], lead plates, bitumen-based membranes, polyethylene [49], and polyester-based membranes." This is dramatic stuff, however your ref. re. steel is slightly incorrect as the authors cited concluded that: "The groundwater level should be checked; if the level is likely to be higher than where the blocking of capillary suction is planned, the blocking should be watertight, e.g. not steel plates". This proviso applies here to all the materials suggested and needs to be built into any engineering works as proposed (but which you do not assess).

Of course what you propose is damaging to the cultural heritage you are trying to protect - as Doehne et al. note - ie. wire sawing out double rows of bricks and mortar at 2 different heights is very invasive as well as expensive and not without risk.

However, even though you suggest replacing the jointing with impermeable materials and old bricks with new more dense impermeable bricks fired at 1100 degrees C (p23 pdf) you also suggest the use of sacrificial bricks which I think sits oddly with the idea of the impermeable double layer, no? Why is this necessary?

So I'm not convinced you have thought this through quite enough from an engineering or conservation perspective and think these suggestions need revisiting and given more thought and properly phased in experimental regimes.

Furthermore you should be at least quantifying whether the extant pointing is impermeable or not - and if so first setting up a test area and replacing it with softer lime mortar to observe if it mitigate the effects of any decay.

You should also be proposing to set up other test areas to trial French drains, both with and without colour-matched sacrificial lime coats and with and without replacing the extant pointing with softer lime mortars ie. all standard practices. Only after these have failed to mitigate the effects of the rising damp should tests then look to more dramatic interventions, but also made first in controlled test sections.

Obviously desalination as proposed in the article to take place once your impermeable barrier has been installed but it had to be thorough with poultice materials well matched to the substrate and salts being treated so this also needs more assessment than apparent in the article.

All of this is well within the scope of a section on further research and I suggest this as a way out of otherwise appearing to have rushed into unsubstantiated claims (p6 pdf) about having successfully 'safeguarding' the blue bricks.

Response 14:

Thank you very much. The author very appreciate this valuable comment. The purpose of this section is to discuss the possible approaches that might be useful for the protection of the heritage site based on the field observation and the methods in other literature. But the field testing part of the effectiveness is not included in this paper. Following your comment, the author rewrote this section thoroughly.

Page 20-22

“7.3 Discussion on anti-weathering strategies

The present protective covering is observed to be made by cement, clay and plant fibres (Fig. 11) in Ping Yao. However, these covering materials failed to retard the weathering of the brick masonry. In reality, it is unsuitable to cover masonry surfaces with waterproofing materials (e.g., impermeable plasters and bitumen) [42]. The misuse of repair materials may itself damage the cultural relics, and is considered improper artificial repair.

Field investigation and weathering analyses have proven that damp transport is the major environmental factor threatening the preservation of brick buildings. Damp transport is directly related to the destructive effects of temperature fluctuations and salt crystallization. Therefore, the guideline of alleviating weathering effects is to decrease the amount and height

of rising damp and to reduce the impact of salt crystallization. To achieve this objective, methods of blocking the rising damp, desalination, and brick replacing should be combined. The properties of blue bricks themselves also should be considered. Studies have confirmed that the firing temperature during brick-making determines the development of the new mineral phases and pore structures in bricks, thus determining the durability [43,44]. To evaluate the influence of the firing temperature on the Ping Yao brick, the authors fired adobes made from Ping Yao clay at 700, 800, 900, 1000, and 1100 °C and tested their properties (Table 6, Table 7).

In order to reduce the influence of damp, one step is to reduce the moisture supply, another step is to increase the moisture evaporation at the near ground area such that the height within the reach rising damp can be limited. Researches conducted by Guimarães et al. (2010) [45] and Ahmad and Rahman (2010) [46] have proven that the ventilation is a effective way to reduce the rising height of damp. Fig. 12 presents a possible method to be implemented: (a) the soil near the walls should be replaced by pebbles as the pebbles cannot transport damp; and (b) a water collection pipe should be place at the lower side of the ditches. This design is expected to serve three purposes: the first is to provide a drainage system and thus reduce the water amount that infiltrated into soils during rainy seasons, the second is to cut the horizontal moisture transportation path and reduce the moisture source, and the third is to affiliate the “natural ventilation” in the area where the moisture concentrates. Considering that the salt crystallization is a main factor in brick weathering, the use of sacrificial bricks as protective layers is suggested. According to the results in Table 6 and Table 7, it is preferable to select bricks fired at 700 °C as sacrificial substrates to trap higher amounts of salts [5,47] due to the higher porosity and higher percent of small pores.

Besides the general methods proposed above, special treatments may be also needed in areas that are severely affected by the weathering phenomena. Desalination methods are

suggested to retard the weathering rate of walls colonized by salt crystals. To remove salt crystals in the brick walls, starch grafted acrylamide [48] and Westox Cocoon poultice [49], are recommended desalination materials due to their effectiveness. For brick replacement, bricks fired at different temperatures should be used as the circumstances may require. Based on the results in Table 6 and Table 7, bricks fired at 1100 °C should be used in replacing the load-bearing parts for their low porosity, favorable PSD, low water absorption and saturation coefficient, high strength, and pore structures beneficial to salt resistance.

It also noteworthy that the synergetic anti-weathering methods discussed above are based on the analyses of laboratory experimental results, the methods reported in literature, and the current condition of the Ping Yao City observed during site investigation. The effectiveness of the proposed methods and subsequent modifications are yet to be studied on the field-testing bases, which is the next step of this study.”

Responses to Reviewer #2:

Comments to the Author(s)

Overall this is an interesting study, addressing a relatively little-researched brick type. The authors have gathered a lot of data across various methods. However, there are issues with this manuscript, in particular in the methodology and associated conclusions, which make it unsuitable for publication in its current state.

The authors mention three different sample types, but throughout the manuscript it is not made clear which samples are analysed. While the reader can work it out with some effort, it would be easier if the authors provided a clear overview of which samples was used for which test and how these findings are tied together. As it stands the study presented here is rather

fragmented. This therefore undermines the credibility of the conservation suggestions made by the authors.

The methodology is not always clearly explained and justified, which again makes it more difficult to follow the narrative laid out by the authors. In fact, some methodologies, such as the repeated heating of blocks, are not justified at all which undermine the interpretations set out in the discussions. Towards the end of the manuscript there are quite a lot of frustrating statements, especially when the authors introduce environmental conditions without providing any sound environmental assessments

P4

The authors need to embed further literature in their introduction. The second paragraph in particular, which describes the firing process for blue bricks, does not contain any further references, and lacks in detail such as firing temperature etc. There does seem to be considerable overlap between traditional red clay brick production (as seen widely around Europe) and blue brick production, yet the authors do not clearly differentiate this. Why not use more of the available literature on red bricks?

P5

L7 – 18 this paragraph is very anecdotal – give more examples and theoretical grounding.

L21- P6L14 There are two issues with this paragraph: [1] the examples cited are predominantly on European red brick, yet the authors have already stated that the blue brick is different – this has not been worked into the narrative presented here and [2] the studies are cited but there is no summary of their findings. So it is not clear how useful these studies are in providing the theoretical framework for the study presented here.

P6

Where are the samples in Fig 2 located in Fig 1?

L18 – 47 Again, there is little citation of the wider literature. It is also unclear what the authors mean with ‘this study not only safeguards brick masonry...’

P7

L15: This needs a reference

L31: Define ‘great preservational value’ – are the authors talking about the value placed on the heritage by the local community?

L59: What evidence do the authors have for the height of the wetting zone?

P8

L5-7: This is a rather vague statement; do these classifications refer directly to the blue brick or is this simply the general list provided by ICOMOS?

L25: ‘The samples belong to two sub-types according to the sampling parts:’ – it is unclear what the authors mean by this

L21-33: The sampling of the bricks needs to be explained in much more detail. It is unclear where the samples were taken, how they were selected and how representative they are. The authors have already flagged up the importance of rising damp, yet do not provide detail on the location of the samples within the wider wall.

L35 – 38: How were these samples selected and obtained?

L41 – 55: The origins and potential weathering history of these bricks seem dubious. Do the authors have further information about their production history and origins?

P9:

L13: Why are they indispensable? Make sure that statements like these are properly explained.

L16: Which samples are the authors referring to? Sample set 1 or 3?

P10:

L12 – 18: Why do the authors not take the elasticity of the material into account when considering the potential damage of frost action?

L 24 – 32: Sample size and drying methods are not justified.

L44: What is ‘pure water’? Distilled or deionised water?

L39 – 57: Why did the authors change the sample size from the one used for the mercury intrusion test? What were the time intervals and how were they determined?

P11:

L8: The use of the word ‘inevitable’ is very odd – the authors really need to be far more thorough in justifying all methods.

L13: Why 0.5 MPa/s? What does this load represent?

L18: This is a very small sample size compared to the original size of the bricks. How do the authors translate these findings to the behaviour of the bricks in situ?

L25 – 60: This whole section is poorly justified. There seems to be no correlation between the cyclical / repeated firing and the production of the original bricks. The authors acknowledge that firing leads to changes in the brick’s mineral phases, yet this is not taken into account during repeated incremental firing. This repeated firing could have altered the bricks sufficiently to influence any further testing.

P12

L25: Why these salts? Are these the ones found in situ?

L28-41: Again, the methods are poorly justified here – why the 2 hour cooling phase? Why not 1 hour or 12 hours? What salts were used in the solution? (this is not clear from the preceding paragraph). There is no indication of use of control samples.

L46 – 58: What is ‘pure water’? Which samples were used? What would the implications of their potential weathering history be?

P13:

L 31-33: 'Gypsum is the product...atmospheres' It is unclear what the function of this sentence is here.

L39-47: The heterogeneity of the bricks stated here undermines the validity of the general conclusions drawn until now, and the general conclusions of the study. The authors need to address this issue clearly.

P14

L7: The heterogeneity of the bricks is not translated in the results presented in table 3.

L14: What are 'ordinary rocks'? L14 – 24 Make little sense to the reader.

L27 – 38: The authors have switched back to general observations, yet do not address the variability of the bricks!

L56-59 / Table 4: The results presented in the text and in the table are not correlated well – it is unclear how the (large!) % ranges presented in the text relate to the results presented in the table.

P15

L4: Which sample?

L20-24: The authors report that the bricks contain anywhere between 22-59% small pores in total connected porosity, which would fall well within the bimodal pore size distribution category. They therefore exclude their own study samples here!

L54: The classifications make little sense when presented here in the text.

P16

L10: Again, this statement ignores the heterogeneity of the bricks

L14: What constitutes an 'ordinary' or 'even porous' rock?

L19: Rising damp is strongly connected with temperatures, and thus evaporation rates. It is not just a function of the capillarity of the bricks. The authors need to take into account the

environmental conditions of the sample sites, and also provide actual measurements to justify their claims of capillary rise heights.

L36 – 52: The authors give the impression here that their samples are drawn from a varied pool of bricks, with large variability in the original material as well as production method. It is therefore difficult to justify their interpretations considering the limited sampling they have undertaken and the variability in the nature of the samples.

P17

L4-27: See my previous point regarding cyclical and incremental testing vs the actual firing method used in the production of these bricks.

L36: What is classified as ‘damage’? Small cracks or structural failure? Pitting of the surface? The graphs do not clearly indicate when the sample is damaged – is it when the line stops?

L48 – 58: Can the authors relate the variability in porosity to the response of the block to the salt weathering?

P18:

L14: In all samples? Are the results pooled? This again ignores the variability in the samples.

L24 – 31: Which samples? The authors are very vague in their statements here

L43-48: This is not clear from the presented results – the freeze thaw tests indicate macro-cracking. The authors do not provide detailed observations of salt weathering across the study site, nor do they provide sufficient information on diurnal temperature fluctuations throughout the year to discard freeze thaw weathering.

L58 – 60: This is the first time the authors introduce clay cores in the walls, which changes the potential behaviour of the bricks.

P19 and P20

This is the first time the authors take environmental conditions into account, and provide little in the way of measurements and observations of justify their statements. The study should be rewritten and restructured with environmental conditions taken into account for the authors to justify making such statements.

P20

L18 – 43: Freeze thaw damage effectiveness is dependent on repeated above and below zero temperatures, more so than moisture content. The authors have not measured temperatures. Therefore, their statements regarding the importance of freeze thaw are not justified.

L46: Frost heave is not the same as frost-induced cracking. Please consult the literature on ‘frost heave’

P21

L3-20: The authors invalidate their own findings, and admit the issues that I have already flagged up in this review. So, how valid are the conclusions drawn from this study? In my view they are not justified.

In light of the problems flagged up regarding the variability of the material in the test results and the discrepancy between in situ conditions, environmental conditions and the laboratory samples it is very questionable whether or not the ‘anti-weathering strategies’ are justified. In my view they are not.

Comment 1:

Comments to the Author(s)

Overall this is an interesting study, addressing a relatively little-researched brick type. The authors have gathered a lot of data across various methods. However, there are issues

with this manuscript, in particular in the methodology and associated conclusions, which make it unsuitable for publication in its current state.

The authors mention three different sample types, but throughout the manuscript it is not made clear which samples are analysed. While the reader can work it out with some effort, it would be easier if the authors provided a clear overview of which samples was used for which test and how these findings are tied together. As it stands the study presented here is rather fragmented. This therefore undermines the credibility of the conservation suggestions made by the authors.

The methodology is not always clearly explained and justified, which again makes it more difficult to follow the narrative laid out by the authors. In fact, some methodologies, such as the repeated heating of blocks, are not justified at all which undermine the interpretations set out in the discussions. Towards the end of the manuscript there are quite a lot of frustrating statements, especially when the authors introduce environmental conditions without providing any sound environmental assessments

Response 1:

Thank you very much for reviewing our manuscript! The authors appreciate your comments and insightful suggestions to our manuscript.

To respond your concern about the methodology and environmental conditions, the authors enriched the more detailed information and the subsequent discussion. For example, the authors added more information about the principle of the magnetic susceptibility and citation; added the max/min value of environmental parameters and related/specified the effects of environmental conditions in discussion sections. Several examples of modification is listed as follows.

Page 5-6

“The environmental information provided by the Ping Yao Meteorological Bureau

indicates that the average annual insolation reaches 2433.2 hours. The average annual temperature is 10.4 °C, ranging from -5.4 °C in January to 24.2 °C in July. The temperature generally below 0 °C from the Mid-October to the Mid-April next year. The average rainfall is 415.5 mm, falling mainly in the summer months. The rainfall from July to September accounts for 58.8 % of total annual rainfall amount, while the rainfall from December to February only accounts for 3.5 % of the total annual rainfall. The annual relative humidity is 58 %. The average wind speed is 2.2 m/s, being higher in spring and winter and lower in summer and autumn. The prevailing wind is mainly from the southwest but from the northeast between June and September.”

Page 18-19

“The brick samples are all damaged after 10 freeze-thaw cycles, which denotes that the freeze-thaw cycles can contribute to the weathering of the Ping Yao ancient bricks. Frost damage could possibly be a combination of three principal theories [41]: the volumetric expansion (approximately 9 %) of water on freezing, the crystallization or "ice lens theory", and the hydraulic pressure theory. The freeze-thaw cycles are definitely detrimental to the preservation of the ancient brick from the laboratorial results. However, whether the freeze-thaw cycles are the dominant factor or not still need further study, since several inconsistencies are observed between the field observation and the laboratory results, as discussed below:

(a) Even though the lab experiment show significant damage caused by freeze-thaw cycles, the effect is not necessarily strong because of the low moisture content of the brick walls during cold season. To assess the influence of damp on the bricks, an IMKO HD2-Mobile Moisture Metre equipped with TRIME-PICO probes was used to measure the moisture content (MC) of the joint mortars between bricks. The MC tests were carried out at

four different locations on June 20th (in the rainy season), 2013. In each location, 4 to 10 different spots ranging from 0.5-1.5 m above ground level were tested. The tests were repeated in the same locations on October 26th (in the dry season), 2013. On June 20th, the maximum MCs (%) of the four measurement locations were 28.49, 65.94, 27.82, and 35.94, respectively, with a mean value of 39.66. However, on October 26th, the corresponding data were reduced to 6.10, 15.30, 7.52, and 10.89, respectively, with a mean value of 9.95 (a 75 % drop). The tested results are consistent with the weather conditions of the Ancient City of Ping Yao, i.e., the average rainfall in the winter only accounts for 3.5 % of the average annual rainfall of 415.5 mm in the county. **Therefore, it seems that there is no enough moisture in bricks during cold to form ice in micro pores.**”

Comment 2:

P4

The authors need to embed further literature in their introduction. The second paragraph in particular, which describes the firing process for blue bricks, does not contain any further references, and lacks in detail such as firing temperature etc. There does seem to be considerable overlap between traditional red clay brick production (as seen widely around Europe) and blue brick production, yet the authors do not clearly differentiate this. Why not use more of the available literature on red bricks?

Response 2:

Thank you very much for your suggestions. According to your suggestions, the authors rewrote this part and added more information from literature including the firing temperatures of the red bricks. The authors also pointed out the two major differences (i.e. presence of oxygen, firing temperatures) between blue and red bricks.

“To create Chinese blue bricks, clay and water need to be blended in exactly the correct proportions, formed into cuboid adobe structures, and dried under natural conditions. Then, the adobes need to be fired in a sealed kiln to form the final products. The fuel used in the kiln is biomass fuel, such as wheat bran, fire wood or charcoal. When oxygen is abundant during the firing process, Fe^{2+} in the abodes will be oxidized into Fe^{3+} , which gives the bricks a red or orange colour. When oxygen is insufficient or when water is added during the latter stage of the firing process, the biomass fuel and water will produce carbon monoxide or hydrogen, both of which can reduce Fe^{3+} into Fe^{2+} , thereby forming a blue brick. The firing temperatures, which directly effect the textural evolution, vitrification, pore size distribution (PSD) of the bricks, and hence the physico-mechanical properties and the weathering resistance, also show some differences between red and blue bricks according to the literature [1,3-5,7]). For example, Elert et al. [5] studied the influence of firing temperature on bricks made by two typical clays, and found that the low-temperature-fired (700-800 °C) generally present inferior PSD, strength, and weathering resistance including freeze-thaw and salt crystallization, compared with the high-temperature-fired (over 1000 °C) bricks. Lopez-Arce et al. [4] found that the firing temperature is directly related to bricks’ properties by controlling the phase changes of minerals, e.g., the new mineral phases (gehlenite and diopside) were formed at the expense of calcite and dolomite above 800 °C. Cultrone et al. [7] found the bricks used for construction of ‘Triangul Bastion’, Riga (Latvia) were fired at around 900 °C, they also found the difference in color corresponds to difference in minerals: yellow color bricks were characterized by quartz, diopside and minor amounts of feldspar; the red bricks were much richer in quartz and feldspar. The data regarding firing temperatures of Chinese blue bricks are limited in literature, Shu et al. [1] studied fourteen blue and red brick samples derived from four modern buildings (constructed between 1866 and 1935) in

Shanghai, China and reported that the maximum firing temperatures of blue bricks are ranging from 600 to 800 °C. The distinctions of brick-making environments (i.e. oxygen, temperature) should produce the blue bricks unique characteristics that are worth of investigation.”

Comment 3:

P5

L7 – 18 this paragraph is very anecdotal – give more examples and theoretical grounding.

L21- P6L14 There are two issues with this paragraph: [1] the examples cited are predominantly on European red brick, yet the authors have already stated that the blue brick is different – this has not been worked into the narrative presented here and [2] the studies are cited but there is no summary of their findings. So it is not clear how useful these studies are in providing the theoretical framework for the study presented here.

Response 3:

Thank you for this useful comment. Indeed, in this section the authors didn't make the framework clear. The authors reorganized this content and specified the point that the fact that red and blue bricks are different but the focus has been mainly on red bricks.

Page 2-3

“Most existing studies have focused on red bricks and have achieved considerable results regarding the influence of the weathering factors and mechanisms such as water impregnation during rainy seasons and water evaporation during dry seasons [2], the raw clay mineral compositions and the mineralogical and textural evolution during the firing process [3-5], the physical and mechanical properties [6], as well as the weathering behaviour [7]. Compared with the widely studied red bricks, the scientific understanding to the blue bricks haven't

receive enough attention, even though the brick-making methods of the traditional Chinese blue brick are very different from the methods used in making European red bricks and the properties may thus be different [1]. ”

Comment 4:

P6

Where are the samples in Fig 2 located in Fig 1?

L18 – 47 Again, there is little citation of the wider literature. It is also unclear what the authors mean with ‘this study not only safeguards brick masonry...’

Response 4:

Thanks for your comment. The authors updated with more detailed information and relevant citations in this section, for the revised version see the response 2 and response 3.

The authors apologize for making this bold yet inappropriate claim. This sentence has been changed as follow.

Page 4

“The objective of this paper is to analyse and discuss the weathering factors of blue bricks and to propose preventive conservation methods. Since there are many historic Chinese architectural structures constructed of blue brick, this study not only **provides understandings and possible approaches on the conversation of** brick masonry in the Ancient City of Ping Yao, which is of great importance, but also relates the possible implications of those results to other similar cases.”

Comment 5:

P7

L15: This needs a reference

L31: Define ‘great preservational value’ – are the authors talking about the value placed on the heritage by the local community?

Response 5:

Thanks for your comment. The preservational value here means the value of the private houses themselves, since these houses are built in hundred of years ago and a part of this UNESCO area. The authors specified this information in context to make it clearer.

Page 5

“The city wall (10 m high and 6.1 km long) has a square layout with the central axis 10-15 ° north by west. This ancient city with six gates contains 4000 ancient private houses, 400 of these houses have great preservational value.”

Comment 6:

L59: What evidence do the authors have for the height of the wetting zone?

Response 6:

Thanks for the comment. The height of the wetting zone is according to the field observation. The height of wetting zone in Fig. 11a is an example.

Page 49

“

Fig. 11 The practice of (a) smearing cement or (b) clay and plant fibres on the ancient brick in walls failed to prevent weathering of the ancient bricks.”

Comment 7:

P8

L5-7: This is a rather vague statement; do these classifications refer directly to the blue brick or is this simply the general list provided by ICOMOS?

Response 7:

Thanks for comment. The ICOMOS is giving a classification based on the appearance of the masonry weathering phenomenon and brief description of the causes. Even though the ICOMOS is mainly about the weathering of rocks, the concept is also can be used for the brick masonry and various types of stone masonry, since they share similar basic principles. For example, the “coving” in ICOMOS is the erosion starting from the interfaces between parts because they have different mineralogy, physical and mechanical properties. Similar effects can also be observed in Fig.2a between the mortar and bricks.

Comment 8:

L25: ‘The samples belong to two sub-types according to the sampling parts:’ – it is unclear what the authors mean by this

Response 8:

Thanks for the comment. To make the sampling part clear, the authors revised the whole section as follow.

Page 6-7

“Samples used in the study came from multiple locations of the Ancient City of Ping Yao (Table 1). Three types of samples were collected from the different locations of the Ancient City:

(a) Weathered ancient brick samples, numbered PY-1 to PY-30, were dispersively taken from the walls of the private houses and the city walls. The weights of the samples range from 10 g to 50 g. The samples can be divided into two sub-types: soft, highly weathered powder samples from the concave parts of the bricks (yellow ellipses in Fig. 2); and hard, slightly weathered fragment samples from the convex parts of the bricks (red circles in Fig. 2). These samples were originally located within the influence range of rising damp (normally at around 0.5 - 1.0 m above the ground). The objective of this test was to investigate the possible causes of different weathering resistance in different part of the same brick from perspectives of mineralogy. The mineral compositions of the samples were therefore determined.

(b) Salt efflorescence samples, numbered S-1 to S-3, were taken from the walls of private houses. The salt efflorescence samples were mainly used to determine the salt crystal types in the Ping Yao ancient city. Since different types of salt have different expansibility, thus the detrimental weathering effects to the brick are variant. Mineral compositions of the samples were determined.

(c) Intact blue bricks were collected from five different locations in Ping Yao and were numbered “i-j”, where “i” denotes the sampling location and “j” denotes the sequential

number of samples at each location. The sampling location 1 to 5 denotes location of “Yard No. 24, Xihujing street”, “Luji Alley”, “Duanzhai Minsu Inn, Yard No. 21, Xinbu Alley, Beihai street”, “Yunzhou Inn, Yard No. 2, Jiaochang street”, “west city wall and east city wall”, respectively. These bricks were collected from restoration or reconstruction sites and were comparatively well preserved. Having had the surface removed, the bricks were cut into $4 \times 4 \times 4$ cm³ cubic samples and numbered “i-j-k”, where “k” denotes the sequential number of cubes cut out of each brick. The cubic samples are termed as *compact* samples in this paper. These samples were used for the physical, mechanical, maximum firing temperatures, salt crystallization and freeze-thaw tests. Samples used to determine pore size distributions (PSD) were drilled out of these cubes.”

Comment 9:

L21-33: The sampling of the bricks needs to be explained in much more detail. It is unclear where the samples were taking, how they were selected and how representative they are. The authors have already flagged up the importance of rising damp, yet do not provide detail on the location of the samples within the wider wall.

L35 – 38: How were these samples selected and obtained?

L41 – 55: The origins and potential weathering history of these bricks seem dubious. Do the authors have further information about their production history and origins?

Response 9:

Thanks for the comments. See response 8.

Comment 10:

P9:

L13: Why are they indispensable? Make sure that statements like these are properly explained.

L16: Which samples are the authors referring to? Sample set 1 or 3?

Response 10:

Thanks for the comments. They are indispensable because the chemical properties, physical properties, and weathering resistance of masonry materials are controlled by the mineralogical constituents. The detailed information can be found in reference [9]. The information of samples and relevant testing items are listed in Table 1 in detail.

Comment 11:

P10:

L12 – 18: Why do the authors not take the elasticity of the material into account when considering the potential damage of frost action?

L 24 – 32: Sample size and drying methods are not justified.

L44: What is ‘pure water’? Distilled or deionised water?

L39 – 57: Why did the authors change the sample size from the one used for the mercury intrusion test? What were the time intervals and how were they determined?

Response 11:

Thanks for comment. The responds to above four questions are listed as follows:

Because it is the tensile force due to icing of water that lead to the breakage of pore walls. When considering the tensile strength, the materials like brick and stone only have very limited elasticity. Such materials have longer elastic range when subjecting to compression, but not when subjecting to tension.

The sample size is listed in Table 1 in detail. The drying method is also included: “After 48 hours of immersion, the samples were taken out to freeze at -20 °C for 12 hours and then thawed in pure water at 20 °C for 2 hours.”

Pure water means deionized water here.

The authors changed the size of samples because the MIP test has its limitation in size of samples (less than 10 mm in diameter and less than 15 mm in height).

Comment 12:

P11:

L8: The use of the word ‘inevitable’ is very odd – the authors really need to be far more thorough in justifying all methods.

L13: Why 0.5 MPa/s? What does this load represent?

L18: This is a very small sample size compared to the original size of the bricks. How do the authors translate these findings to the behaviour of the bricks in situ?

Response 12:

Thanks for your comments. According to your comment, the authors deleted this word.

The loading rate is according to the national standard in China, a reference is added, as listed in the following box.

Indeed, the sample size is smaller than the original size, because the limitation of number of samples allowed by local government. The authors also agree with the fact that the strength obtained by smaller samples and by big samples has some differences due to size effect. But it is still very good to have the change to get the ancient samples and conduct this test to see the strength.

“The UCS was carried out in a TAW-2000 electric-fluid servo-controlled testing system at a loading rate of 0.5 MPa/s (GB/T 50266-2013) [16].”

Comment 13:

L25 – 60: This whole section is poorly justified. There seems to be no correlation between the cyclical / repeated firing and the production of the original bricks. The authors acknowledge that firing leads to changes in the brick’s mineral phases, yet this is not taken into account during repeated incremental firing. This repeated firing could have altered the bricks sufficiently to influence any further testing.

Response 13:

Thanks for the comment. The authors are following the method described in references [1] and [17], in which contain the detailed information and deeper interpretation and validation of this technique. It would be too long if the authors include the whole explanation of this technique in this paper.

Comment 14:

P12

L25: Why these salts? Are these the ones found in situ?

L28-41: Again, the methods are poorly justified here – why the 2 hour cooling phase? Why not 1 hour or 12 hours? What salts were used in the solution? (this is not clear from the preceding paragraph). There is no indication of use of control samples.

L46 – 58: What is ‘pure water’? Which samples were used? What would the implications of their potential weathering history be?

Response 14:

Thanks for these comments. The authors selected these salts for two main reasons, the first is that these salts are observed in salt efflorescence in Ping Yao (as described in page 12), and the second is these three salts has different expansibility during crystallization. And the pure water means the deionized water, the authors changed this word to avoid misunderstanding. The samples used are included in Table 1 in detail.

Comment 15:

P13:

L 31-33: ‘Gypsum is the product...atmospheres’ It is unclear what the function of this sentence is here.

Response 15:

Thank you for your comments. According to your comments, the authors revised the content, specified the reaction process of the formation of gypsum.

Page 12

“Gypsum is the product of the reaction of sulfur compounds in polluted atmospheres (e.g. $\text{Ca}^{2+} + \text{SO}_2 \rightarrow \text{CaSO}_4$) [24,25].”

Comment 16:

L39-47: The heterogeneity of the bricks stated here undermines the validity of the general conclusions drawn until now, and the general conclusions of the study. The authors need to address this issue clearly.

P14

L7: The heterogeneity of the bricks is not translated in the results presented in table 3.

L14: What are ‘ordinary rocks’? L14 – 24 Make little sense to the reader.

L27 – 38: The authors have switched back to general observations, yet do not address the variability of the bricks!

L56-59 / Table 4: The results presented in the text and in the table are not correlated well – it is unclear how the (large!) % ranges presented in the text relate to the results presented in the table.

Response 16:

Thanks for your comments. The authors had the same concern of effects of heterogeneity of bricks while conducting the basic physical parameters. It, of course, would be better if the whole brick can be tested in different direction (e.g. the water absorption in three direction), however, due to the limitation of the samples, the authors were looking for an alternative way, that is, testing many cubes and using standard deviation to reflect the variance of the data (Table 3).

The authors added the a references defining “ordinary rocks” and “porous rock”: [27] Von Moos A., De Quervin F..1948. Technische Gesteinskunde, Birkhäuser, Basel.

The percentage in context is calculated from the data listed in Table 4. The reason why the authors present the real value of intrusion volume in stead of the percentage in Table 4 is because the data in Table 4 are also used to calculate the SSI, which cannot be calculated in percentage form.

Comment 17:

P15

L4: Which sample?

L20-24: The authors report that the bricks contain anywhere between 22-59% small pores in total connected porosity, which would fall well within the bimodal pore size distribution category. They therefore exclude their own study samples here!

L54: The classifications make little sense when presented here in the text.

Response 17:

Thanks for commenting. The samples for each test are listed in Table 1 in detail.

The connected porosity is calculated based on the PSD data, which not the related to the bimodal of the PSD at all. The bimodal and unimodal is the shape of the PSD curve that representing the two types of pore distributions. Generally, unimodal is the most pores are mainly near one pore size, while the bimodal is the pores are concentrating on two sizes (e.g. inter and intra aggregates pores is bimodal). In our case, the PSD curve are unimodal, as shown in Fig. 3.

Page 41

Fig. 3 Demonstration of the PSD curve of an compact sample derived from Ping Yao ancient brick.”

Comment 18:

P16

L10: Again, this statement ignores the heterogeneity of the bricks

L14: What constitutes an ‘ordinary’ or ‘even porous’ rock?

Response 18:

See response 16.

Comment 19:

L19: Rising damp is strongly connected with temperatures, and thus evaporation rates. It is not just a function of the capillarity of the bricks. The authors need to take into account the environmental conditions of the sample sites, and also provide actual measurements to justify their claims of capillary rise heights.

Response 19:

Thank you very much for this comment. The author agree with this opinion that the rising damp also a function of environmental conditions such as temperature and evaporation rates. In this paper, the author mainly focus on the effects of the properties of the bricks itself. The detailed investigation of the effects of environmental factor is another focus.

Comment 20:

L36 – 52: The authors give the impression here that their samples are drawn from a varied pool of bricks, with large variability in the original material as well as production method. It is therefore difficult to justify their interpretations considering the limited sampling they have undertaken and the variability in the nature of the samples.

Response 20:

Thanks for your comment. The samples used in this study are indeed collected from the various locations in Ping Yao City and the magnetic susceptibility test also indicated that the bricks were fired at two sets of temperatures. However, the material used for brick making was from the same location. The authors collected the samples from various site is because this paper is to study the Ping Yao blue bricks as a whole to get a bigger picture. The variance of data is reflected in the tables.

Comment 21:

P17

L4-27: See my previous point regarding cyclical and incremental testing vs the actual firing method used in the production of these bricks.

Response 21:

Thanks for your comment. The authors conducted the test following the technique proposed in literature [26], in which has the detailed description about the principle and validation of this technique.

Comment 22:

L36: What is classified as ‘damage’? Small cracks or structural failure? Pitting of the surface? The graphs do not clearly indicate when the sample is damaged – is it when the line stops?

L48 – 58: Can the authors relate the variability in porosity to the response of the block to the salt weathering?

P18:

L14: In all samples? Are the results pooled? This again ignores the variability in the samples.

L24 – 31: Which samples? The authors are very vague in their statements here

L43-48: This is not clear from the presented results – the freeze thaw tests indicate macro-cracking. The authors do not provide detailed observations of salt weathering across the study site, nor do they provide sufficient information on diurnal temperature fluctuations throughout the year to discard freeze thaw weathering.

Response 22:

Thanks for the comments. According to your comments and suggestions given by another reviewer, the authors have rewritten the whole section, the new content is included in the following box.

Page 18-21

“The brick samples are all damaged after 10 freeze-thaw cycles, which denotes that the freeze-thaw cycles can contribute to the weathering of the Ping Yao ancient bricks. Frost damage could possibly be a combination of three principal theories [41]: the volumetric expansion (approximately 9 %) of water on freezing, the crystallization or "ice lens theory", and the hydraulic pressure theory. The freeze-thaw cycles are definitely detrimental to the preservation of the ancient brick from the laboratorial results. However, whether the freeze-thaw cycles are the dominant factor or not still need further study, since several inconsistencies are observed between the field observation and the laboratory results, as discussed below:

(a) Even though the lab experiment show significant damage caused by freeze-thaw cycles, the effect is not necessarily strong because of the low moisture content of the brick walls during cold season. To assess the influence of damp on the bricks, an IMKO HD2-Mobile Moisture Metre equipped with TRIME-PICO probes was used to measure the moisture content (MC) of the joint mortars between bricks. The MC tests were carried out at four different locations on June 20th (in the rainy season), 2013. In each location, 4 to 10 different spots ranging from 0.5-1.5 m above ground level were tested. The tests were repeated in the same locations on October 26th (in the dry season), 2013. On June 20th, the maximum MCs (%) of the four measurement locations were 28.49, 65.94, 27.82, and 35.94, respectively, with a mean value of 39.66. However, on October 26th, the corresponding data

were reduced to 6.10, 15.30, 7.52, and 10.89, respectively, with a mean value of 9.95 (a 75 % drop). The tested results are consistent with the weather conditions of the Ancient City of Ping Yao, i.e., the average rainfall in the winter only accounts for 3.5 % of the average annual rainfall of 415.5 mm in the county. Therefore, it seems that there is not enough moisture in bricks during cold to form ice in micro pores.

(b) The frost heaving damage modes in the experiment (Fig. 9) are not quite consistent with the modes observed on-site. In the laboratory, the typical frost heaving damage mode of the ancient bricks is macroscopic cracking, and the cleavages of different samples are at roughly equal levels (Fig. 9). The failure modes observed on-site include contour scaling and granular disintegration, which are more similar to the failure modes caused by salt crystallization; while the flaking observed on site is commonly a product of freeze-thaw weathering.

Therefore, it can be confirmed that the freeze-thaw is a factor accounting for the weathering of ancient blue brick in Ping Yao. Yet the questions of “how detrimental is the freeze-thaw” and “is the freeze-thaw a more or less dominant weathering factor compared with the salt crystallization” are still unclear and need to be further investigated in next step. To have a quantitative understanding, the moisture content in brick walls, rainfall condition, and underground water condition in wet and dry seasons need to be monitored at more locations; and then the lab-based freeze-thaw cycles should be conducted on the samples with certain moisture contents that based on the field results.”

Comment 23:

L58 – 60: This is the first time the authors introduce clay cores in the walls, which changes the potential behaviour of the bricks.

Response 23:

Thanks for comment. The presence of clay core is mainly affecting the damp rising process and the amount of soluble salts (see reference [26]). Indeed, the presence of clay cores has influence to the behavior of bricks such as stability. But in this paper, the main focus is about the relationship between the properties of blue brick and the weathering resistance. Therefore, the presence of clay core doesn't bring too much differences.

Comment 24:

P19 and P20

This is the first time the authors take environmental conditions into account, and provide little in the way of measurements and observations of justify their statements. The study should be rewritten and restructured with environmental conditions taken into account for the authors to justify making such statements.

Response 24:

Thank you for your comment. According to your comment, the authors rewritten the introduction parts about the environment condition and the whole part regarding the effects of freeze-thaw cycles, as listed in the following box.

Page 5-6

“The environmental information provided by the Ping Yao Meteorological Bureau indicates that the average annual insolation reaches 2433.2 hours. The average annual temperature is 10.4 °C, ranging from -5.4 °C in January to 24.2 °C in July. The temperature generally below 0 °C from the Mid-October to the Mid-April next year. The average rainfall

is 415.5 mm, falling mainly in the summer months. The rainfall from July to September accounts for 58.8 % of total annual rainfall amount, while the rainfall from December to February only accounts for 3.5 % of the total annual rainfall. The annual relative humidity is 58 %. The average wind speed is 2.2 m/s, being higher in spring and winter and lower in summer and autumn. The prevailing wind is mainly from the southwest but from the northeast between June and September.”

Page 18-20

“7.2 Effect of the freeze-thaw cycles

... The freeze-thaw cycles are definitely detrimental to the preservation of the ancient brick from the laboratorial results. However, whether the freeze-thaw cycles are the dominant factor or not still need further study, since several inconsistencies are observed between the field observation and the laboratory results, as discussed below:

(a) Even though the lab experiment show significant damage caused by freeze-thaw cycles, the effect is not necessarily strong because of the low moisture content of the brick walls during cold season. To assess the influence of damp on the bricks, an IMKO HD2-Mobile Moisture Metre equipped with TRIME-PICO probes was used to measure the moisture content (MC) of the joint mortars between bricks. The MC tests were carried out at four different locations on June 20th (in the rainy season), 2013. In each location, 4 to 10 different spots ranging from 0.5-1.5 m above ground level were tested. The tests were repeated in the same locations on October 26th (in the dry season), 2013. On June 20th, the maximum MCs (%) of the four measurement locations were 28.49, 65.94, 27.82, and 35.94, respectively, with a mean value of 39.66. However, on October 26th, the corresponding data were reduced to 6.10, 15.30, 7.52, and 10.89, respectively, with a mean value of 9.95 (a 75 %

drop). The tested results are consistent with the weather conditions of the Ancient City of Ping Yao, i.e., the average rainfall in the winter only accounts for 3.5 % of the average annual rainfall of 415.5 mm in the county. Therefore, it seems that there is no enough moisture in bricks during cold to form ice in micro pores.

(b) The frost heaving damage modes in the experiment (Fig. 9) are not quite consistent with the modes observed on-site. In the laboratory, the typical frost heaving damage mode of the ancient bricks is macroscopic cracking, and the cleavages of different samples are at roughly equal levels (Fig. 9). The failure modes observed on-site include contour scaling and granular disintegration, which are more similar to the failure modes caused by salt crystallization; while the flaking observed on site is commonly a product of freeze-thaw weathering.

Therefore, it can be confirmed that the freeze-thaw is a factor accounting for the weathering of ancient blue brick in Ping Yao. Yet the questions of “how detrimental is the freeze-thaw” and “is the freeze-thaw a more or less dominant weathering factor compared with the salt crystallization” are still unclear and need to be further investigated in next step. To have a quantitative understanding, the moisture content in brick walls, rainfall condition, and underground water condition in wet and dry seasons need to be monitored at more locations; and then the lab-based freeze-thaw cycles should be conducted on the samples with certain moisture contents that based on the field results.”

Comment 25:

P20

L18 – 43: Freeze thaw damage effectiveness is dependent on repeated above and below zero temperatures, more so than moisture content. The authors have not measured temperatures. Therefore, their statements regarding the importance of freeze thaw are not justified.

L46: Frost heave is not the same as frost-induced cracking. Please consult the literature on ‘frost heave’

Response 25:

Thanks for the comments. The freeze-thaw test was conducted in accordance with the BS EN 12371 [23] recommendation.

Comment 26:

P21

L3-20: The authors invalidate their own findings, and admit the issues that I have already flagged up in this review. So, how valid are the conclusions drawn from this study? In my view they are not justified.

In light of the problems flagged up regarding the variability of the material in the test results and the discrepancy between in situ conditions, environmental conditions and the laboratory samples it is very questionable whether or not the ‘anti-weathering strategies’ are justified. In my view they are not.

Response 26:

Thank you for the comment. The authors made a major revision according to your comments and suggestions. The anti-weathering strategies are mainly about the discussion of the possible way to reduce the weathering rate based on the laboratory findings and the site conditions. The author agree that the claim of anti-weathering strategy is too bold because there is no data of field testing for these methods in this paper. This part therefore has been

rewritten and the new contents is included in the following box. The effectiveness of different candidate methods will be further studied through field testing.

Page 20-23

“7.3 Discussion on anti-weathering strategies

The present protective covering is observed to be made by cement, clay and plant fires (Fig. 11) in Ping Yao. However, these covering materials failed to retard the weathering of the brick masonry. In reality, it is unsuitable to cover masonry surfaces with waterproofing materials (e.g., impermeable plasters and bitumen) [42]. The misuse of repair materials may itself damage the cultural relics, and is considered improper artificial repair.

Field investigation and weathering analyses have proven that damp transport is the major environmental factor threatening the preservation of brick buildings. Damp transport is directly related to the destructive effects of temperature fluctuations and salt crystallization. Therefore, the guideline of alleviating weathering effects is to decrease the amount and height of rising damp and to reduce the impact of salt crystallization. To achieve this objective, methods of blocking the rising damp, desalination, and brick replacing should be combined. The properties of blue bricks themselves also should be considered. Studies have confirmed that the firing temperature during brick-making determines the development of the new mineral phases and pore structures in bricks, thus determining the durability [43,44]. To evaluate the influence of the firing temperature on the Ping Yao brick, the authors fired adobes made from Ping Yao clay at 700, 800, 900, 1000, and 1100 °C and tested their properties (Table 6, Table 7).

In order to reduce the influence of damp, one step is to reduce the moisture supply, another step is to increase the moisture evaporation at the near ground area such that the height within the reach rising damp can be limited. Researches conducted by Guimarães et al.

(2010) [45] and Ahmad and Rahman (2010) [46] have proven that the ventilation is a effective way to reduce the rising height of damp. Fig. 12 presents a possible method to be implemented: (a) the soil near the walls should be replaced by pebbles as the pebbles cannot transport damp; and (b) a water collection pipe should be place at the lower side of the ditches. This design is expected to serve three purposes: the first is to provide a drainage system and thus reduce the water amount that infiltrated into soils during rainy seasons, the second is to cut the horizontal moisture transportation path and reduce the moisture source, and the third is to affiliate the “natural ventilation” in the area where the moisture concentrates. Considering that the salt crystallization is a main factor in brick weathering, the use of sacrificial bricks as protective layers is suggested. According to the results in Table 6 and Table 7, it is preferable to select bricks fired at 700 °C as sacrificial substrates to trap higher amounts of salts [5,47] due to the higher porosity and higher percent of small pores.

Besides the general methods proposed above, special treatments may be also needed in areas that are severely affected by the weathering phenomena. Desalination methods are suggested to retard the weathering rate of walls colonized by salt crystals. To remove salt crystals in the brick walls, starch grafted acrylamide [48] and Westox Cocoon poultice [49], are recommended desalination materials due to their effectiveness. For brick replacement, bricks fired at different temperatures should be used as the circumstances may require. Based on the results in Table 6 and Table 7, bricks fired at 1100 °C should be used in replacing the load-bearing parts for their low porosity, favorable PSD, low water absorption and saturation coefficient, high strength, and pore structures beneficial to salt resistance.

It also noteworthy that the synergetic anti-weathering methods discussed above are based on the analyses of laboratory experimental results, the methods reported in literature, and the current condition of the Ping Yao City observed during site investigation. The effectiveness of the proposed methods and subsequent modifications are yet to be studied on the field-testing

bases, which is the next step of this study.”

The authors are deeply grateful to the anonymous reviewers for helpful comments!

Appendix A

Responses to Reviewers' Comments

Zhongjian Zhang

China University of Geosciences, Beijing, China.

zhangzhongjian@cugb.edu.cn

May, 2020

Dear Editor and Reviewer,

We would like to submit our revised research manuscript entitled “*Characteristics and weathering mechanisms of the traditional Chinese blue brick from the Ancient City of Ping Yao*” (ID: **RSOS-200058**) for your consideration for publication in the journal *Royal Society Open Science*.

We have made a point-by-point response to the reviewer’s comments and suggestions, including a detailed description of any requested or suggested revisions.

We have also carefully checked and corrected the writing format and errors to make our revised manuscript conform to the journal style.

All the modifications and explanations in this revised version are listed in detail in the following “**Responses to Reviewers' Comments**”.

We would deeply appreciate your consideration and reviewer’s helpful comments and suggestions.

Yours Sincerely,

Zhongjian Zhang, Ph.D

Department of Civil Engineering, School of Engineering and Technology, China University of Geosciences, Beijing, China.

E-mail address: zhangzhongjian@cugb.edu.cn, Phone number: 86-10-82322627, Fax number: 86-10-82322624.

Acknowledgement The authors are grateful to the anonymous reviewer for a careful checking of the details and for helpful comments that improved this paper.

Responses to Reviewer #1:

Comment 1:

Thank you for responding to many of the comments I previously made.

Although you have moderated your claims I do think you still need to strongly recast the work as part of a larger study ie. as a pilot project or phase 1 of a study to describe the deterioration mechanisms affecting the blue bricks, rather than assert here without sufficient data that salts are the only major factor.

Response 1:

Thank you very much for reviewing our manuscript! The authors appreciate your detailed comments and insightful suggestions to our manuscript. The present study indeed is a phase 1 study of our ongoing project. The indoor water uptake experiments and in-situ field monitoring is still running and will be reported after obtaining satisfying results. This paper is a conclusion of completed element tests.

According to your suggestions, the authors made a revision. The details are included in following sections of this response file. Note that contents marked in red color are contents modified during this revision. Contents marked in blue are contents that had been revised in previous revision.

In the meanwhile, the authors also pointed out that this paper is phase 1 of our study as shown in following box:

“This paper presents results of the phase one work of our study. The objective of this paper is to analyse and discuss the weathering factors of blue bricks and to propose preventive conservation methods based on lab experiments and field condition, and to provide a relative narrow range of methods to be further tested in fields. ”

Page 22 in Conclusions

“This paper presents results of the phase one study of our research project. Through on-site investigation and laboratory tests, the weathering types, physico-mechanical properties, and resistance to salt crystallization and freeze-thaw cycles of the bricks in the Ancient City of Ping Yao were studied.”

Comment 2:

This is because there are still 2 important problems:

1) Both I and the other reviewer have misgivings about sample size, non-homogeneity of the material composition vs. homogeneity of samples, and the environmental testing/data used (or not used) etc, which impact the assertion that salt crystallization is the major deterioration factor for the bricks (cf. 7.1).

It may well be the case that salts are the biggest factor but you have not given enough weight to the effects of freeze-thaw action which, by the now added figures from the local Meteorology Bureau, suggest that freezing temperature conditions occur for 6 months (<0 centigrade October-April) and by extrapolating from the Bureau's rainfall figures, significant to moderately significant rain occurs for at least 3 months that overlap those freezing temperatures (October, November and March).

The fact that water uptake by the bricks is reported and that rising damp is observed up to 3m, and that the PSD profile benefits both salt crystallization and freeze/thaw, all suggest

freeze/thaw needs further investigating as you acknowledge. This means the assertions in 7.1 etc need modifying.

Response 2:

Thank you very much for your suggestions. Indeed, although the authors think that effects of salt crystallization is more detrimental than freeze-thaw cycles, the overlapping of freezing and rainfall periods needs to be considered and present data is not able to support this assertion. A figure showing daily temperature and rainfall/snowfall condition is also included in the supplementary material to give a more precise data support. More investigation and data are needed to judge if the previous assertions in section 7.1 was true or not. Therefore, the authors modified section 7.1 and related parts. The authors think that this comment is also an insightful suggestion to our research project - to study the separate and coupling effects of salt solutions and freeze-thaw cycles by applying salt concentration and temperature conditions similar to research area in environment chamber.

The revised contents are listed as following box:

Page 2 in abstract:

“Salt crystallization and freeze-thaw cycles were found to be two important factors that lead to brick weathering.”

Page 17

“7.1 Effect of salt crystallization

The weathering types and the basic properties of bricks (e.g. porosity, water absorption, PSD and SSI, and salt crystallization test) indicate that salt crystallization is an important contributor to brick damage [39-40]. A schematic representation of the weathering process of the brick walls in the Ancient City of Ping Yao is shown in Fig. 10.”

“7.2 Effect of the freeze-thaw cycles

The brick samples are all damaged after 10 freeze-thaw cycles, which denotes that **the freeze-thaw cycles is another contributor to the weathering of the Ping Yao ancient bricks.** Frost damage could possibly be a combination of three principal theories [42]: the volumetric expansion (approximately 9 %) of water on freezing, the crystallization or "ice lens theory", and the hydraulic pressure theory. Even though **the laboratorial results indicate that the freeze-thaw cycles are detrimental to the preservation of the ancient brick, several inconsistencies still remain in this pioneer study of ancient brick weathering, as discussed below:**

(a) To assess seasonal influence of damp on the brick walls, an IMKO HD2-Mobile Moisture Metre equipped with TRIME-PICO probes was used to measure the moisture content (MC) of the joint mortars between bricks. The MC tests were carried out at four different locations on June 20th (in the rainy season), 2013. In each location, 4 to 10 different spots ranging from 0.5-1.5 m above ground level were tested. The tests were repeated in the same locations on October 26th (in the dry season), 2013. On June 20th, the maximum MCs (%) of the four measurement locations were 28.49, 65.94, 27.82, and 35.94, respectively. On October 26th, the corresponding data were reduced to 6.10, 15.30, 7.52, and 10.89, respectively. **Although the rainfall/snowfall amount is small in winter, there are rainfall periods that overlap freezing temperatures (i.e. October, November, and March) in cold seasons. Existence of small amount of water in walls may still cause damage to bricks due to brick's PSD profile that benefits effects of freeze-thaw.**

(b) On the other hand, the lab experiments show significant damage caused by freeze-

thaw cycles, but the frost heaving damage modes in the experiment (Fig. 9) are not quite consistent with the modes observed on-site. In the laboratory, the typical frost heaving damage mode of the ancient bricks is macroscopic cracking, and the cleavages of different samples are at roughly equal levels (Fig. 9). The failure modes observed on-site include contour scaling and granular disintegration, which are more similar to the failure modes caused by salt crystallization.

Therefore, it can be confirmed that the freeze-thaw is a factor accounting for the weathering of ancient blue brick in Ping Yao. However, according to present information, whether the contribution of freeze-thaw cycles outweigh that of salt crystallization to the weathering is still unclear and need to be further investigated in next step. To have a quantitative understanding, the moisture content in brick walls, rainfall condition, and underground water condition in wet and dry seasons need to be monitored at more locations; and then the lab-based freeze-thaw cycles should be conducted on the samples with certain moisture contents that based on the field results.

Page 20

“Damp transport is directly related to the destructive effects of salt crystallization and temperature fluctuations (i.e. freeze-thaw cycles). Therefore, the guideline of alleviating weathering effects is to decrease the amount and height of rising damp and to reduce the impact of salt crystallization and freeze-thaw. ”

Page 23-24

“(6) Salt crystallization and freeze-thaw cycle are two important factors accounting for the brick weathering of Ping Yao ancient bricks. The damp risen from the foundation soil evaporates into the atmosphere under the effect of wind or/and sunlight, leaving soluble salts stranded on the brick surface and forming crystals. The pressure generated in the process can

cause brick damage. The freeze-thaw cycles also lead to brick damage due to overlapping of rainfall/snowfall and freezing periods. A field monitoring scheme is suggested for further study on the effects of the freeze-thaw.”

Comment 3:

2) The 2nd point is that the 'coving' described is attributed to gypsum formation/solubility and/or mineral inhomogeneity in the bricks themselves.

However, although the 'brick-mortar' interface is mentioned as contributing there is no description of what the mortar actually is. I did ask for this to be reported in my first review.

Fig 2 c & d shows what appears to be a white mastic as mortar/pointing (old mastic are often an oil mixed with CaCO_3 eg. chalk) and this seems to be proud of the surface suggesting that the coving is determined by water egress from the brick faces and not joints. This means that any soluble salts will of course egress out from the brick faces too, rather than the joints. So I am not convinced the authors have fully described the cause of the coving here and I suggest they look at the permeability of the mortar compared with the brick. There is much literature on this kind of scaling and its relation to bedding/pointing (especially with regard to stone decay).

Response 3:

Thanks for your comment. The authors apologize for not mentioning the mortar type. The mortar in Pingyao is mainly lime or soil-lime mixture. Organic additive is only added in important parts such as city walls, because Pingyao City is located in arid inland instead of rich coastal region. The type of organic additive used in Pingyao was a thin gruel made from sticky rice and water. The authors have also tested many samples collected from different locations using iodine, two samples collected from the city wall of Pingyao is shown in

pictures below (Note: three samples are collected from different locations of city walls, the samples in plastic bags are after iodine treatment). In fact, the lime mortars in Ming Dynasty and Qing dynasty of China were mainly using sticky rice as additives, which is also reported in literature (Liu et al, 2016. <http://dx.doi.org/10.1080/15583058.2015.1104399>).

About deeper explanation of “coving” in brick-mortar interface. Even though the obtained mortar samples are too small to conduct direct permeability test in labs available for the authors. But the authors found an alternative method to evaluate it. The MIP method was used to determine the PSD and open porosity of the mortar used in Pingyao (as shown in supplementary materials). Results show that the mortar has smaller open porosity than brick in general, the PSD curve also indicate that majority of pores in mortar also has smaller pore radii compared with bricks. The above two points lead to a conclusion that mortar has smaller permeability compared with the ancient brick. Therefore, the “coving” effect between brick and mortar is very likely an important factor to weathering, as you mentioned. Thank you for this comment, again. The authors also revised relevant contents accordingly, as shown in following box:

Page 12

“The inhomogeneities in mineral composition may be a factor that results in different weathering resistances both among different bricks and also in the different parts of the same brick (Fig. 2) [26].

Salt efflorescence on ancient bricks are mainly composed of hexahydrate ($\text{MgSO}_4 \cdot 6\text{H}_2\text{O}$), epsomite ($\text{MgSO}_4 \cdot 7\text{H}_2\text{O}$) and sodium nitrate (NaNO_3). A small amount of gypsum ($\text{CaSO}_4 \cdot 2\text{H}_2\text{O}$) and bloedite ($\text{Na}_2\text{Mg}(\text{SO}_4)_2 \cdot 4\text{H}_2\text{O}$) are also found. Sulfate (e.g., magnesium sulfate and sodium sulfate) often presents strong expandability during crystallization [26].”

Page 13-14

“The porosity and PSD results of bricks also suggest a greater permeability of bricks compared with mortar (lime or soil-lime mixture), since the mortar has smaller open porosity (14.69 %) and the majority of pores are also smaller. In light of the explanation in the ICOMOS-ISCS [8], the erosion of the brick-mortar interface in Fig. 2 is also likely related to the alveolization (or coving) that caused by different permeability between the mortar and the brick.”

Comment 4:

Further points:

In terms of mitigating the effects of both salt and water egress, as I said before testing needs to be made with both softer lime pointing (but being mindful of local traditions) and also lime washes (again mindful of traditions). Lime washes as sacrificial coating in principle push the zone of salt efflorescence further forward from the brick faces and these could also be tested.

Desalination methods also need proper investigating rather than just the assertion of specific products by citation. Poulticing can be appropriate but only if the wall is isolated from the source of soluble salts - otherwise its pointless.

Response 4:

Thank you for your suggestions.

To respond your first suggestion, although the use of lime as sacrificial coating is potential to be an effective protecting method, this method is not recommended in the restoration and preservation practice of ancient bricks in recent years for aesthetic reasons. The new governmental principle regarding historical heritage site requires that the restoration or reconstruction of historical relics cannot change the original appearance. Therefore, the authors put the focus on investigating thin bricks to serve as sacrificial layers. But your suggestion is very good to our further work, we also need to consider the properties of lime/mortar when studying the usage of bricks as sacrificial layers, since the properties of bonding material between thin bricks are also important to the final effectiveness, too.

To respond your second suggestion. The authors strongly agree with it. It is indeed inappropriate to directly suggest the application of these products just because previous literature. A product tested effective in one place is not necessarily working well in another place. Therefore, the authors modified the content and suggested to use them as first batch of products to be tested. The lateral damp blocking and ventilation are also suggested to work together with desalination methods. Although the proposed damp blocking method is not as good as putting a membrane into the walls which some people worried about the stability of walls, theoretically it can also isolate quite amount damp transportation and reduce the damp transportation. Of course, we still need to wait for monitoring data to give a percentage. The revised contents are listed in the following box.

“Besides the general methods proposed above, special treatments may be also needed in areas that are severely affected by the weathering phenomena. Desalination methods are suggested to retard the weathering rate of walls colonized by salt crystals. According to previous cases reported in literature, starch grafted acrylamide [49] and Westox Cocoon poultice [41] are recommended as the first batch of desalination materials to conduct field testing in order to evaluate the effectiveness and suitability for salt removing in Ping Yao brick walls. ”

The authors are deeply grateful to the anonymous reviewers for helpful comments!

Appendix C

Cover Letter for Revised Manuscript

July, 2020

Dear Editor and Reviewer,

Thank you for accepting our research manuscript entitled “*Characteristics and weathering mechanisms of the traditional Chinese blue brick from the Ancient City of Ping Yao*” (ID:RSOS-200058.R1) for publication in the journal *Royal Society Open Science*. We would like to submit the revised version of our manuscript.

We have made a point-by-point response to the reviewers’ comments and suggestions, including a detailed description of any requested or suggested revisions.

We have also carefully checked and corrected the writing format and errors to make our revised manuscript conform to the journal style. The language is also polished with the help of a language editing company called American Journal Experts.

All the modifications and explanations in this revised version are listed in detail in the following “**Responses to Reviewers' Comments**”.

Yours Sincerely,

Zhongjian Zhang, Ph.D

Department of Civil Engineering, School of Engineering and Technology, China
University of Geosciences, Beijing, China.

E-mail address: zhangzhongjian@cugb.edu.cn, Phone number: 86-10-82322627, Fax
number: 86-10-82322624.